# Questioning the Coverage-Length Metric in Conformal Prediction: When Shorter Intervals Are Not Better

Yizhou Min [* 1]   Yizhou Lu [* 2]   Lanqi Li [* 1]   Zhen Zhang [1]   Jiaye Teng [† 1 3]

## Abstract

Conformal prediction (CP) has become a cornerstone of distribution-free uncertainty quantification, conventionally evaluated by its coverage and interval length. This work critically examines the sufficiency of these standard metrics. We demonstrate that *the interval length might be deceptively improved through a counter-intuitive approach* termed Prejudicial Trick (PT), while the coverage remains valid. Specifically, for any given test sample, PT probabilistically returns an interval, which is either null or constructed using an adjusted confidence level, thereby preserving marginal coverage. While PT potentially yields a deceptively lower interval length, it introduces practical vulnerabilities: the same input can yield completely different prediction intervals across repeated runs of the algorithm. We formally derive the conditions under which PT achieves these misleading improvements and provide extensive empirical evidence across various regression and classification tasks. Furthermore, we introduce a new metric *interval stability* which helps detect whether a new CP method implicitly improves the length based on such PT-like techniques. Code is available at https://github.com/benben-cd/PT-Conformal-Prediction

## 1. Introduction

Machine learning has been successfully applied in numerous fields. However, machine learning models often suffer from overconfidence issues (Guo et al., 2017; Minderer

et al., 2021), making them unreliable for deployment in high-stakes areas such as medicine and finance (De Prado, 2018). Therefore, it is crucial to develop techniques for uncertainty quantification and calibrate the original model to enhance the reliability of predictions (Sullivan, 2015; Minderer et al., 2021; Smith, 2024).

Among all the uncertainty quantification methods, conformal prediction (CP) stands out due to its simplicity and distribution-free characteristics (Vovk et al., 2005; Shafer & Vovk, 2008; Angelopoulos & Bates, 2021). CP is a post hoc approach for constructing prediction intervals, based on a non-conformity score calculated on a hold-out calibration set (Algorithm 2). CP and its variants have demonstrated promising performances in numerous applications (Lei & Candès, 2021; Angelopoulos et al., 2022).

Generally, researchers evaluate the intervals returned by CP via two criteria: *coverage* and *interval length*. Firstly, a valid coverage ensures that the actual response value has a high probability of falling within the interval. Secondly, the interval is encouraged to be as short as possible, as a shorter interval provides more precise information about prediction uncertainties. These two evaluation metrics are commonly used in the literature (Tibshirani et al., 2019; Teng et al.; Angelopoulos et al., 2023; He & Lam, 2024) and a branch of works improves the length with several different meaningful approaches (Romano et al., 2019; Izbicki et al., 2020; Teng et al.; Guan, 2023; Stutz et al., 2022). This raises a question on the potential risk of evaluating the CP methods narrowly on these standard metrics:

> **The key question:**
>
> If a new algorithm outperforms existing ones on coverage and length, should it automatically be considered superior for practical deployment?

Specifically, can a CP method maintain valid coverage and *deceptively improve interval length metrics* through counter-intuitive constructions, while *introducing practical risks*? Notably, this differs from the existing literature that proposes a new method that falls short in the length metric while performing better in the new metric (*e.g.*, conditional coverage). To illustrate how it happens, we next consider the following Example 1:

---
*Equal contribution  [1]School of Statistics and Data Science, Shanghai University of Finance and Economics, Shanghai, China [2]Department of Statistics and Data Science, Fudan University, Shanghai, China [3]Institute of Data Science and Statistics, Shanghai University of Finance and Economics, Shanghai, China. Correspondence to: Jiaye Teng <tengjiaye@sufe.edu.cn>.

*Proceedings of the 43rd International Conference on Machine Learning*, Seoul, South Korea. PMLR 306, 2026. Copyright 2026 by the author(s).

---

**Algorithm 1** Prejudicial Trick (PT)

---
1: **Input:** conformal prediction algorithm (base) $\mathcal{A}_{1-\alpha}(\cdot; \hat{\mu})$, test point $\boldsymbol{x}'$, probability $p$.
2: Generate a uniform random variable $U \sim \text{Unif}([0,1])$;
3: **if** $U > p$: **then**
4:     Interval $\mathcal{C}_{1-\alpha}(\boldsymbol{x}') = [\hat{\mu}(\boldsymbol{x}'), \hat{\mu}(\boldsymbol{x}')]$ (regression tasks) or $\mathcal{C}_{1-\alpha}(\boldsymbol{x}') = \varnothing$ (classification tasks);
5: **else**
6:     Calculate the adjusted miscoverage rate $\alpha' = 1 - \frac{1-\alpha}{p}$;
7:     Interval $\mathcal{C}_{1-\alpha}(\boldsymbol{x}') = \mathcal{A}_{1-\alpha'}(\boldsymbol{x}'; \hat{\mu})$;
8: **end if**
9: **Output:** Interval $\mathcal{C}_{1-\alpha}(\boldsymbol{x}')$.

---

*Example* 1 (The Pitfalls of Length.). Two doctors, Alice and Bob, are estimating recovery time for patients after treatment. CP with historical data reveals that $60\%$ of patients recover within 4 years, and $80\%$ within 5 years. When a new patient asks for an estimated recovery time, Alice and Bob adopt distinct strategies:

- Alice: Assign recovery time interval $[0, 4]$ years;
- Bob: Assign recovery time interval $[0, 5]$ with probability 0.75, while $[0, 0]$ with probability 0.25.

For both strategies in Example 1, $60\%$ of patients fall in the estimated interval in expectation, thus satisfying the criteria for valid marginal coverage. Besides, Bob's approach yields a shorter average interval length $5 \times 75\% = 3.75 \ll 4$. Overall, Bob achieves a shorter interval while achieving the same coverage as Alice. However, Bob's strategy is flawed in its practical application, since (a) from the micro-level, Bob provides different intervals for the same patient if queried multiple times, and (b) from the macro-level, Bob randomly informs $25\%$ of patients that they will recover immediately after treatment regardless of their actual condition. The example is illustrated in Figure 3.

In this paper, inspired by the motivating example (Example 1), the *Prejudicial Trick* (PT) emerges as a practically invalid method that artificially shortens prediction intervals in CP (see Algorithm 1 and Figure 4). Instead of providing consistent intervals, PT assigns null intervals with a fixed probability to any test sample, and assigns confidence intervals with lower miscoverage rates in other cases to maintain the marginal coverage. While PT preserves marginal coverage and potentially reduces the average interval length, its rationale is less sound compared to standard methods like Vanilla Conformal Prediction (VCP). Specifically, PT suffers from two limitations:

- *Instability issues*: Repeated runs of PT produce different intervals for the same input;
- *Unfairness issues*: PT provides informative predictions for only a subset of test samples, while assigning unin-

formative null intervals to the rest[1].

From the theoretical perspective, we offer several theoretical results to provide deep understandings of PT regarding both coverage and length, and informally summarize them in Theorem 1.1.

**Theorem 1.1** (Theoretical Results Summary). *We term* base *as the base CP algorithm, term* PT *as the base algorithm with PT, and omit mild assumptions for clarity.* **For coverage**, *it holds that,*

- *PT satisfies marginal coverage guarantees and further inherits conditional coverage guarantees of base CP (Theorem 3.3);*
- *PT outperforms its base regarding the conditional coverage under some conditions, even if the base does not satisfy conditional coverage guarantees (Remark 3.5).*

***For length**, it holds that:*

- *PT achieves shorter average intervals than its base under some general conditions (Lemma 3.6);*
- *We provide sufficient conditions under which PT reduces the average interval length for both differentiable (Theorem 3.7) and non-differentiable (Theorem 3.10) length functions. Notably, these conditions are often satisfied in the common scenario of model misspecification (Remark 3.11).*
- *These results lead to corollaries for specific cases, such as when the length function is locally concave (Corollary 3.8) or when the base algorithm is VCP (Corollary 3.9).*
- *We also provide a success case (Example 3) and failure case where PT cannot decrease the average length (Example 4).*

From the experimental perspective, we verify our findings on various real-world datasets, regarding marginal and conditional coverage (Figure 1), and interval length (Table 2). Besides, we validate our findings under different settings, including different tasks (classification regimes in Table 4) and other CP algorithms (Conformalized Quantile Regression in Table 5).

However, the improvement on length is vacuous, as discussed in Section 3.5. To make this failure mode explicit, we further introduce *Interval Stability*, a complementary diagnostic that quantifies the run-to-run variation of the prediction set for the same input. Rather than replacing coverage, length, or downstream utility-based evaluation, *Interval Stability* helps contextualize reported length improvements when the returned prediction sets depend on stochastic components of the algorithm (Remark 4.3).

*Remark* 1.2 (PT Hacks the Coverage-Length Metric). We

---

[1]In this paper, unfairness stems from the fact that a portion of samples are assigned null intervals in a run, even though each has an equal probability of being prejudiced. This differs from the unfairness concept grounded in conditional coverage (Zhao et al., 2020), where individuals are prejudiced based on their features.

present PT not as a practical solution, despite its theoretical advantages on the coverage-length metric. Instead, it serves as a **cautionary example that highlights issues such as instability and unfairness** during deployment. PT represents a simple way to *hack* the coverage-length metric. This reminds us of identifying where the actual improvements come from within a potentially complex algorithm, since they may implicitly contain PT-like tricks.

*Remark* 1.3 (Practical Relevance of the PT with Current CP Methods). PT clarifies a possible risk in stochastic CP pipelines that when the returned prediction set depends on randomized components, average coverage and average length alone may hide run-to-run variation for the same input. For example, localized CP (Hore & Barber, 2024) often relies on an estimated local scale function, which may vary across training runs when learned by randomized procedures such as neural networks or random forests. Proposition 3.2 shows that PT can be viewed as a degenerate limiting case of such scale-based normalization. This connection is not a criticism of localized CP itself but a motivation of introducing stability-related diagnostics whenever randomness in model training or score construction can affect the returned prediction sets. Another example is in classification task where randomized tie-breaking procedures are used in methods such as APS/RAPS (Romano et al., 2020; Angelopoulos et al., 2020) which may also introduce mild algorithmic randomness. Our point is not that such procedures are PT-like, but that their stochasticity should be made explicit when interpreting coverage and length. We empirically revisit these issues in Section 4 by comparing interval stability for deterministic baselines and stochastic CP pipelines.

*Remark* 1.4 (Why Study PT In Conformal Prediction). The relevance between PT and CP lies in the model misspecification (Remark 3.11). Specifically, model misspecification serves as a potential sufficient condition where PT works, and it generally appears in real-world applications of CP. However, the existence of model misspecification does not always hold for other interval estimation tasks, and therefore limits the extensions of PT beyond CP.

## 2. Related Work

Conformal prediction (Vovk et al., 2005) is mainly evaluated by the coverage-length metric. For coverage, CP provides finite sample guarantees under exchangeability assumptions (Vovk et al., 2005; Tibshirani et al., 2019; Barber et al., 2023), ensuring that prediction sets achieve the expected marginal coverage. Another related metric is conditional coverage, which is unachievable in finite sample settings without further assumptions (Vovk, 2012). Therefore, recent work focuses on various relaxations of conditional coverage (Foygel Barber et al., 2021; Gibbs et al., 2025).

Another metric is the average *interval length*, as shorter inter-

vals are generally more informative (Lei et al., 2018; Sadinle et al., 2018). Numerous methods aim to construct adaptive intervals that reduce length while maintaining valid coverage. One line of work designs alternative non-conformity score functions: for example, Romano et al. (2019) integrate quantile regression, while Guan (2023) propose a localized method that adapts to test-time information. Other score functions have been proposed in (Feldman et al., 2021; Alaa et al., 2023; Han et al., 2022; Teng et al.). Of particular interest is the method of Izbicki et al. (2020), which estimates the asymptotic conditional distribution of the non-conformity scores and constructs prediction intervals based on high-density regions. Another line of work involves a training procedure for interval-length optimization, such as CPL (Kiyani et al., 2024), CP-Gen (Bai et al., 2022), ConfTr (Stutz et al., 2022), and BoostedCP (Xie et al., 2024). Different from existing methods that provide principled and meaningful advances in CP, our proposed prejudicial trick achieves efficiency gains through an illusory construction, making the length improvement non-substantive.

Besides coverage and length, several auxiliary metrics have been introduced. These include *excess* and *deficit* (Seedat et al., 2023), which measures the extent to which the prediction intervals are unnecessarily wide or insufficiently narrow; *false positive rate* (Fisch et al., 2022), which improves precision by limiting the number of incorrect labels in classification settings; and *conditional weighted coverage* (Jensen et al., 2024)—a hybrid metric that takes both coverage and length into account. Among them, a particularly important evaluation criterion is *group coverage* (Cauchois et al., 2021), which assesses the coverage and interval length across population subgroups defined by features or response magnitudes. However, in practice, people still tend to prioritize the coverage and length metrics (Lei et al., 2018; Cresswell et al., 2024; Zhang et al., 2024; Xu et al., 2025). Notably, this paper introduces a new metric distinct from existing approaches. Unlike existing studies that design metrics mainly to showcase favorable performance but often struggle with length, we show that PT potentially surpasses its base model in length while performing poorly under the new metric. We provide additional related works on CP and interval regression in Appendix A.2.

## 3. Prejudicial Trick with Deceptive Improvement

In this section, we challenge the coverage-length metric in CP by constructing a trick in Section 3.2. Specifically, this trick potentially improves the interval length while maintaining the coverage, yet it introduces instability and unfairness issues. We further investigate how this trick influences the coverage (Section 3.3) and the length (Section 3.4) theoretically and empirically. Finally, we discuss more details on

*Table 1.* Results for the synthetic datasets (motivating example in Section 3.2). Comparison between VCP and PT-VCP under different $\alpha$ levels, regarding length and coverage.

| $\alpha$ | $p$ | VCP | | PT-VCP | |
|---|---|---|---|---|---|
| | | Coverage | Length | Coverage | Length |
| 0.10 | 0.96 | $0.906 \pm 0.004$ | $22.894 \pm 0.138$ | $0.909 \pm 0.005$ | $\mathbf{22.614} \pm 0.254$ |
| | 0.98 | $0.906 \pm 0.004$ | $22.894 \pm 0.138$ | $0.904 \pm 0.004$ | $\mathbf{22.714} \pm 0.165$ |
| 0.20 | 0.96 | $0.792 \pm 0.011$ | $21.886 \pm 0.125$ | $0.799 \pm 0.011$ | $\mathbf{21.255} \pm 0.136$ |
| | 0.98 | $0.792 \pm 0.011$ | $21.886 \pm 0.125$ | $0.796 \pm 0.010$ | $\mathbf{21.589} \pm 0.149$ |

the deceptive improvements of this trick in Section 3.5.

### 3.1. Preliminary

**Conformal Prediction.** CP creates statistically rigorous uncertainty sets for any predictive model. Given $\boldsymbol{X}$ as the input and $\alpha \in (0, 1)$ as the miscoverage rate, CP returns an uncertainty set[2] $\mathcal{C}_{1-\alpha}(\boldsymbol{X})$ that satisfies

$$\mathbb{P}(y \in \mathcal{C}_{1-\alpha}(\boldsymbol{X})) \geq 1 - \alpha, \tag{1}$$

where $y$ denotes the true response of feature $\boldsymbol{X}$. We omit the detailed discussion in Appendix A.4.

**Notations.** Let $\{Z_i\}_{i=1}^n$ denote $n$ i.i.d. samples drawn from the distribution $\mathcal{P}_Z$, where $Z \in \mathbb{R}$. Denote $\{Z_{(i)}\}_{i=1}^n$ as the order statistics of $\{Z_i\}_{i=1}^n$ arranged in decreasing order, *i.e.*, $Z_{(1)} \geq Z_{(2)} \geq \cdots \geq Z_{(n)}$. The empirical $\tau$-th quantile with $n$ samples is defined as $\hat{Q}_\tau(\{Z_i\}_{i=1}^n) := Z_{(\lceil (n+1)(1-\tau) \rceil)}$. Let $\varnothing$ denote the empty set, and $\mathbb{I}(\cdot)$ denote the indicator function. For a given set $\mathcal{C}$, let $|\mathcal{C}|$ denote the measure of the set. In this paper, we denote a trained CP algorithm that directly outputs the $1 - \alpha$ confidence interval given a test point as $\mathcal{A}_{1-\alpha}(\cdot; \hat{\mu})$ for simplicity, where $\hat{\mu}$ is the machine learning algorithm used in the CP algorithm.

### 3.2. Prejudicial Trick

In this section, we propose a trick (Algorithm 1) used in CP with the intuition from Example 1 and deploy this trick in a motivating example on a synthetic dataset. The experiment results in Table 1 demonstrate that this trick improves the interval length while maintaining the marginal coverage compared to its base. We begin with the construction process of this trick:

**Construction Process.** For each test point, assign a null set[3] with probability $1 - p$, and assign the interval with an adjusted miscoverage rate $\alpha' = 1 - \frac{1-\alpha}{p}$ for the remaining

---

[2] CP algorithms either return a set measured by its size (for classification tasks), or return an interval measured by its length (for regression tasks). We do not distinguish between these terms throughout the paper.

[3] The null set represents a set of measure zero. It can be an empty set, or a single-point set in regression.

$p$ portion of test points, where $p \in (1 - \alpha, 1)$. Overall, for a new test point $\boldsymbol{x}'$, the interval is constructed as Equation (2):

$$\mathcal{C}_{1-\alpha}^{\text{PT}}(\boldsymbol{x}') = \begin{cases} \text{null set} & \text{with the probability } 1 - p, \\ \mathcal{C}_{1-\alpha'}^{\text{CP}}(\boldsymbol{x}') & \text{with the probability } p. \end{cases} \tag{2}$$

We call this process Prejudicial Trick (PT, see in Algorithm 1). Note that PT can be directly applied to any base CP algorithm. To illustrate that PT improves length without sacrificing marginal coverage, we empirically consider a motivating example in Example 2.

*Example* 2 (Synthetic Dataset). *Consider a regression setting where the true underlying model is a linear model with the Gaussian mixture noise, given by $Y = \boldsymbol{X}^\top \boldsymbol{\beta} + \epsilon$, with $\boldsymbol{X} \sim \mathcal{N}(\boldsymbol{0}, I_2)$. The noise term $\epsilon$ follows $\mathcal{N}(\mu, 1)$ with probability $0.5$, and $\mathcal{N}(-\mu, 1)$ with probability $0.5$. The training fold, calibration fold, and test fold are generated based on this underlying distribution. We deploy VCP (Algorithm 2) and PT-VCP under such regimes. We refer to Appendix D.1.1 for more details.*

**Results and Discussions of Example 2.** Tabel 1 illustrates the results of Example 2, where *PT-VCP improves the length while maintaining the marginal coverage compared to VCP*. The results validate that the coverage-length metric could be hacked by invalid tricks like PT. The insights are as follows: the construction of $\epsilon$ guarantees $\mathcal{C}_{1-\alpha'}^{\text{CP}}$ close to $\mathcal{C}_{1-\alpha}^{\text{CP}}$ regarding the length when choosing proper $\alpha$ and $p$. Therefore, PT potentially improves the average length when averaging with those null sets.

Notably, Foygel Barber et al. (2021) propose a method similar to PT. Unlike PT which emphasizes the potential length improvement, Foygel Barber et al. (2021) mainly center on conditional coverage metrics. Moreover, we extend the scope of PT in Remark 3.1 by relaxing the notion of the null set.

*Remark* 3.1 (The extension of PT). PT in Algorithm 1 heavily relies on the notion of null sets. Fortunately, one can extend this null set with an interval returned by CP with a small coverage rate. For example, PT can be constructed as

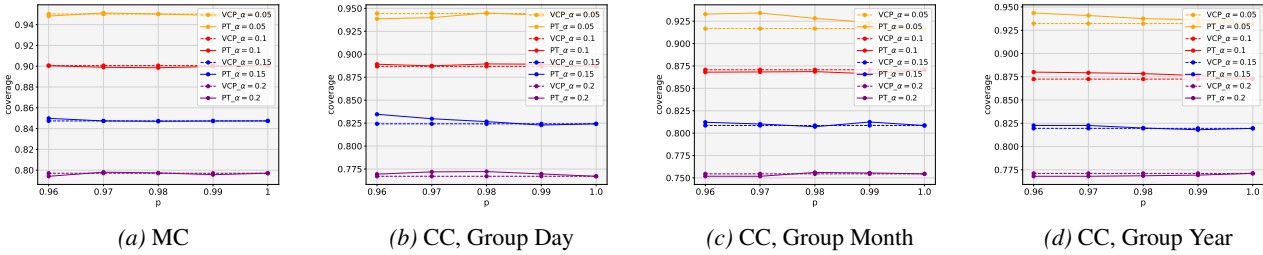

*(a) MC*      *(b) CC, Group Day*      *(c) CC, Group Month*      *(d) CC, Group Year*

*Figure 1.* Comparing the (a) marginal coverage and (b, c, d) conditional coverage between VCP with and without PT. Results demonstrate that PT would not significantly change the marginal coverage; and PT has better conditional coverage compared to the base algorithm.

follows:

$$\mathcal{C}_{1-\alpha}^{\mathrm{PT}}(\boldsymbol{x}') = \begin{cases} \mathcal{C}_{1-\alpha_1'}^{\mathrm{CP}}(\boldsymbol{x}') & \text{with the probability } 1-p, \\ \mathcal{C}_{1-\alpha_2'}^{\mathrm{CP}}(\boldsymbol{x}') & \text{with the probability } p, \end{cases}$$
$$\tag{3}$$

where $(1-p)\alpha_1' + p\alpha_2' = \alpha$ which guarantees the marginal coverage and $\alpha_1'$ is sufficiently small. We mainly consider the null set in this paper to simplify the related discussions.

Although PT appears to be an artificially constructed special case, we find that it strongly correlates with localized CP. We demonstrate that PT is a special case of localized CP in Proposition 3.2.

**Proposition 3.2** (Relation Between PT And Localized CP). *In localized CP, we use the normalized nonconformity score to construct prediction sets, which is defined as*

$$\hat{S}^{norm}(x, y) = \hat{S}(x, y)/\hat{\sigma}(x), \tag{4}$$

*where $\hat{S}(x, y)$ is the nonconformity score and $\hat{\sigma}(\cdot)$ is a trained estimator of local scale. Consider an extreme case that $\hat{\sigma}$ can only output $0^+$ and $\frac{q_{1-\alpha'}}{q_{1-\alpha}}$ with probability $1-p$ and $p$ respectively[4], where $q_{1-\alpha'}$ and $q_{1-\alpha}$ are quantiles on calibration data. In this case, intervals returned by localized CP are equivalent to the form in Equation (2).*

In practice, beyond the two-point distribution in Proposition 3.2, the interval length returned by localized CP across retraining runs might become a random variable following a more general distribution with a lower expectation. Consequently, even if one fixes a single trained $\hat{\sigma}$, retraining $\hat{\sigma}$ multiple times and reporting a favorable run amounts to cherry-picking, yielding shorter length without leveraging task information, while marginal coverage remains valid.

### 3.3. Coverage

This section investigates the coverage guarantee of CP with PT. We prove that PT maintains the marginal coverage and conditional coverage guarantees in Theorem 3.3 and discuss

---

[4]If we train $\hat{\sigma}$ at each run, the randomness during training might lead to different outputs given a fixed $x$.

the conditional coverage of PT when the conditional coverage of base CP is not valid in Remark 3.5. The empirical validation in Figure 1 supports the theoretical findings.

We begin with the formal guarantee of marginal and conditional coverage of PT in Theorem 3.3. We prove that PT keeps marginal coverage and further keeps conditional coverage if the base interval satisfies the conditional guarantee.

**Theorem 3.3** (Coverage Guarantee of PT). *Assume that the exchangeability assumption holds (see Proposition A.3). Then the interval returned by Algorithm 1 with the adjusted miscoverage rate $\alpha' = 1 - \frac{1-\alpha}{p}$ guarantees that*

$$\mathbb{P}(y' \in \mathcal{C}_{1-\alpha}^{PT}(\boldsymbol{X}')) \geq 1 - \alpha, \tag{5}$$

*where $(\boldsymbol{X}', y')$ denotes a new test point. If for any $\alpha$, $\mathbb{P}(y \in \mathcal{C}_{1-\alpha}^{CP}(\boldsymbol{X}') \mid \boldsymbol{X}') \geq 1 - \alpha$ holds for $\boldsymbol{X}'$ almost surely, then*

$$\mathbb{P}(y \in \mathcal{C}_{1-\alpha}^{PT}(\boldsymbol{X}') \mid \boldsymbol{X}') \geq 1 - \alpha \tag{6}$$

*holds for any $\alpha$ and for $\boldsymbol{X}'$ almost surely.*

The key intuition behind Theorem 3.3 is that, for marginal coverage, the null set (with probability $1-p$) and the enlarged interval set with miscoverage $\alpha'$ (with probability $p$) reach the marginal coverage guarantees $p(1-\alpha') = 1-\alpha$; and for conditional coverage, the randomness within PT is independent of the specific input. Therefore, such randomness is averaged out given a specific input, thus keeping the conditional coverage unchanged.

*Remark* 3.4 (Comparison to the Tradeoffs between Conditional Coverage and Interval Length). Existing works on CP have analyzed the potential tradeoffs between conditional coverage and interval length (Foygel Barber et al., 2021; Gibbs et al., 2025). Our work differs from this line, since PT does not operate by creating such a trade-off. Specifically, Theorem 3.3 validates that PT does not violate the conditional coverage whenever the base algorithm satisfies it. See Section 3.5 for further discussions.

The conditional coverage guarantee in Theorem 3.3 requires that the base algorithm satisfies the conditional coverage guarantees. However, this requirement does not always hold

in practice. We next discuss in Remark 3.5 that PT still exhibits the potential to outperform the base algorithm even when the requirement does not hold.

*Remark* 3.5 (When Base CP Violates Conditional Coverage). Theorem 3.3 shows that PT preserves conditional coverage whenever the base algorithm already satisfies it. Without this assumption, the theorem gives no conditional coverage guarantee for PT. However, its conditional coverage can still exceed that of the base algorithm on subset $\mathcal{A}$. Let $\mathbb{P}\big(y \in C_{1-\alpha}^{\mathrm{CP}}(X) \mid X \in \mathcal{A}\big) = 1 - f_{\mathcal{A}}(\alpha)$, where $f_{\mathcal{A}}(\alpha)$ denotes the true conditional miscoverage rate of the base method on $\mathcal{A}$. For PT, let $\alpha' = 1 - (1-\alpha)/p$. For the null-set version of PT, the corresponding subset-level coverage is $\mathbb{P}\big(y \in C_{1-\alpha}^{\mathrm{PT}}(X) \mid X \in \mathcal{A}\big) = p\,(1 - f_{\mathcal{A}}(\alpha'))$. Hence, the coverage difference between PT and the base CP on $\mathcal{A}$ is

$$[f_{\mathcal{A}}(\alpha) - f_{\mathcal{A}}(\alpha')] - (1-p)\,(1 - f_{\mathcal{A}}(\alpha')). \tag{7}$$

Thus, PT exceeds the coverage of base method on $\mathcal{A}$ when the miscoverage reduction from using the more conservative level $\alpha'$ outweighs the coverage loss caused by the null-set.

**Experiments.** We compare marginal and conditional coverage rates returned with and without PT on BIKE dataset (Fanaee-T, 2013) using VCP (Algorithm 2) as the base algorithm. We omit the implementation details here and refer to Appendix D.2.2 for details. The results in Figure 1 demonstrate that (a) PT preserves the marginal coverage (Figure 1a); (b) When VCP fails to guarantee the group coverage, PT-VCP fails as well. Experimental results demonstrate that PT achieves comparable group coverage with its base models (Figure 1b, Figure 1c, Figure 1d). We provide more experiments on group coverage with different real-world datasets in Appendix C.1.

### 3.4. Length

This section investigates the sufficient conditions under which PT improves interval length while keeping the coverage unchanged, and further conducts experiments to validate the theoretical findings. We first propose Lemma 3.6 as a weak sufficient condition. We then derive a more informative condition under both differentiable regimes (Theorem 3.7) and non-differentiable regimes (Theorem 3.10). We further discuss the special cases on the local concave assumption (Corollary 3.8) and VCP regimes (Corollary 3.9), and find that model misspecification provides one practical mechanism through which these sufficient conditions may hold (Remark 3.11). We finally present a failure case in Example 4 when PT cannot outperform its base regarding the interval length.

Experiment results on various datasets in Table 2 align closely with the theoretical results. Besides, we conduct experiments on classification tasks (Table 4), different base algorithms (Table 5), and conduct ablation studies on differ-

ent hyperparameters (Figure 6-Figure 15).

**Additional Notations.** We introduce the following notations to facilitate the discussions in this section. Let $\alpha$ denote the miscoverage rate and $s(\boldsymbol{x}, y; \hat{\mu})$ denote the score function, where $\hat{\mu}(\cdot)$ denotes the learned model. Let $\mathcal{C}_{1-\alpha}^{\mathrm{CP}}(\boldsymbol{x})$ denote the interval returned by CP at point $\boldsymbol{x}$, and $\mathcal{C}_{1-\alpha}^{\mathrm{PT}}(\boldsymbol{x})$ denote the interval returned by its PT-variant (Algorithm 1). Let $\mathcal{L}(\boldsymbol{x}, 1-\alpha; s)$ denote the length of the returned interval at point $\boldsymbol{x}$ with miscoverage $\alpha$, *i.e.*, $|\mathcal{C}_{1-\alpha}^{\mathrm{CP}}(\boldsymbol{x})|$. Furthermore, we denote $\mathcal{G}(u; s) = \mathbb{E}_X(\mathcal{L}(\boldsymbol{X}, u; s))$, where $u$ is coverage level.

We next prove a series of sufficient conditions under which PT improves the interval length, starting from Lemma 3.6 which provides a straightforward sufficient condition.

**Lemma 3.6** (General Sufficient Condition). *If exists* $p \in (1 - \tilde{\alpha}, 1)$ *such that*

$$p\mathcal{G}((1 - \tilde{\alpha})/p; s) < \mathcal{G}(1 - \tilde{\alpha}; s), \tag{8}$$

*where* $\tilde{\alpha}$ *denotes the miscoverage rate. Then the interval length returned by PT (with parameter $p$) outperforms that of its base algorithm,*

$$\mathbb{E}|\mathcal{C}_{1-\tilde{\alpha}}^{PT}(\boldsymbol{X}')| < \mathbb{E}|\mathcal{C}_{1-\tilde{\alpha}}^{CP}(\boldsymbol{X}')|, \tag{9}$$

*where the expectation is taken over the testing point* $\boldsymbol{X}'$.

The intuition behind Lemma 3.6 is pretty simple: PT assigns null sets with a fixed probability whose measure is zero, thus potentially reducing the average length. Although the sufficient condition in Lemma 3.6 is general, the absence of additional assumptions makes it uninformative in practice. To obtain more insights, we next introduce a differentiable assumption in Theorem 3.7.

**Theorem 3.7** (First-order Condition). *Assume $\mathcal{G}$ is first-order differentiable and satisfies*

$$\frac{\mathcal{G}(1 - \tilde{\alpha}; s)}{1 - \tilde{\alpha}} > \left.\frac{\partial}{\partial u}\mathcal{G}(u; s)\right|_{u=1-\tilde{\alpha}}, \tag{10}$$

*where $\tilde{\alpha}$ denotes the miscoverage rate and the expectation is taken over $\boldsymbol{x}$. Then there exists a parameter $p$ in Algorithm 1, such that the interval length returned by PT outperforms that of its base algorithm, namely*

$$\mathbb{E}|\mathcal{C}_{1-\tilde{\alpha}}^{PT}(\boldsymbol{X}')| < \mathbb{E}|\mathcal{C}_{1-\tilde{\alpha}}^{CP}(\boldsymbol{X}')|, \tag{11}$$

*where the expectation is taken over the testing point* $\boldsymbol{X}'$.

Theorem 3.7 follows the insights of Lemma 3.6, and further utilizes Equation 10 as the sufficient condition, which characterizes the local behavior of the interval length function. Theorem 3.7 provides more insights on when and how PT outperforms its base algorithm regarding the length metrics. We next derive a localized concave condition in Corollary 3.8 based on Theorem 3.7.

**Corollary 3.8** (Localized Concave Conditions). *Under the settings in Theorem 3.7, the sufficient condition in Equation 10 holds if $\mathcal{G}(u; s)$ is strictly concave on $u \in [0, 1 - \tilde{\alpha}]$.*

Corollary 3.8 provides a condition under which PT outperforms its base algorithm regarding length. Consider a regression problem with additive noise $y = f^*(x) + \epsilon$. If the noise distribution exhibits local concavity and the base model approximates the true function $f^*$ well, then the length function $\mathcal{G}(u; s)$ generally satisfies the localized concavity property. Consequently, the performance improvement of PT is guaranteed by Corollary 3.8. This inspires the construction of Example 2.

Besides, Corollary 3.9 focuses on the settings of deploying VCP. Since VCP returns the same length for each individual, the expectation operator in Theorem 3.7 degenerates.

**Corollary 3.9** (Deterministic Case). *Under the settings in Theorem 3.7, if applying VCP (Algorithm 2) as the base algorithm, the sufficient condition in Equation 10 holds if*

$$\frac{\mathcal{L}(\boldsymbol{x}, 1 - \tilde{\alpha}; s)}{1 - \tilde{\alpha}} > \left. \frac{\partial}{\partial u} \mathcal{L}(\boldsymbol{x}, u; s) \right|_{u = 1 - \tilde{\alpha}}, \quad (12)$$

*where the expectation operator degenerates due to the characteristics of VCP.*

Unfortunately, real-world applications may not satisfy the differentiability assumption in Theorem 3.7. Therefore, we relax this assumption and obtain the secant condition from Lemma 3.6 directly in Theorem 3.10.

**Theorem 3.10** (Secant Sufficient Condition). *If there exists $u \in (1 - \tilde{\alpha}, 1)$, such that*

$$\frac{\mathcal{G}(1 - \tilde{\alpha}; s)}{1 - \tilde{\alpha}} > \frac{\mathcal{G}(u; s) - \mathcal{G}(1 - \tilde{\alpha}; s)}{u - (1 - \tilde{\alpha})}, \quad (13)$$

*where $\tilde{\alpha}$ denotes the miscoverage rate and the expectation is taken over $\boldsymbol{x}$. Then there exists a parameter $p = (1 - \tilde{\alpha})/u$ in Algorithm 1, such that the interval length returned by PT outperforms its base algorithm, namely,*

$$\mathbb{E}|\mathcal{C}_{1-\tilde{\alpha}}^{PT}(\boldsymbol{X}')| < \mathbb{E}|\mathcal{C}_{1-\tilde{\alpha}}^{CP}(\boldsymbol{X}')|, \quad (14)$$

*where the expectation is taken over the testing point $\boldsymbol{X}'$.*

Theorem 3.10 shares similar intuitions with Theorem 3.7, and further relaxes the differentiability assumption by comparing the secant slopes. Informally, PT achieves smaller average lengths than its base algorithm when the length function does not grow extremely fast within the region $(1 - \tilde{\alpha}, 1)$.

Theorem 3.10 shows that PT reduces average length whenever the averaged length curve grows sublinearly over some interval to the right of the target coverage level. Importantly, this phenomenon is not restricted to discrete prediction sets.

The following Example 3 gives a simple continuous setting where the averaged length function satisfies the condition exactly, and hence PT strictly improves the average length. *Example* 3 (Success Case). If the values of the nonconformity score satisfy $\mathbb{P}(S \leq r) = r^2, \ r \in [0, 1]$, it holds that PT strictly reduces the expected length:

$$\mathbb{E}\left|C_{1-\tilde{\alpha}}^{PT}(X')\right| < \mathbb{E}\left|C_{1-\tilde{\alpha}}^{CP}(X')\right|. \quad (15)$$

*Remark* 3.11 (Relationship Between Misspecification and Sufficient Condition). Model misspecification is a common practical scenario that aligns with our theoretical analysis, as it typically satisfies the sufficient conditions in Theorem 3.7 and Theorem 3.10 (Wang & Blei, 2020; Huang et al., 2023). **Specifically, misspecification could lead to a residual with a non-zero mean, resulting in a non-convex length function. This outcome is closely related to the local concavity condition in Corollary 3.8.** For this reason, we employ misspecification regimes in most of our experiments.

However, the aforementioned sufficient conditions are not always satisfied. We present a failure case in Example 4 and illustrate the empirical validation in Figure 5. *Example* 4 (Failure Case). If the values of the nonconformity score in VCP follow a Gaussian distribution over randomness in $\boldsymbol{x}$, then for all $\alpha \in (0, 1)$, and all $p \in (1 - \alpha, 1)$, it holds that

$$\mathbb{E}|\mathcal{C}_{1-\alpha}^{PT}(\boldsymbol{X}')| > \mathbb{E}|\mathcal{C}_{1-\alpha}^{CP}(\boldsymbol{X}')|. \quad (16)$$

Example 4 demonstrates that PT does not always outperform its base algorithm regarding the coverage-length metric. In this case, the quantile curve grows fast enough near the tail, so increasing the coverage costs more length than the outer factor $p$ can compensate. However, our goal is not to present PT as a universally applicable method, but to demonstrate a risk in the coverage-length evaluation. To achieve this, the existence of any realistic scenarios where PT can create deceptively shorter intervals is sufficient.

**Experiment.** We conduct several experiments comparing the length returned with and without PT using VCP (Algorithm 2) on MEPS19-21 (Cohen et al., 2009), BIKE (Fanaee-T, 2013), BLOG-DATA (Buza, 2014), BIO (Rana, 2013), FACEBOOK1-2 (Singh, 2015), CONCRETE (Yeh, 1998), STAR (Achilles et al., 2008). To simulate model misspecification, we manually add bias to the label (as shown in the bias column in Table 2). We refer to Appendix D.2 for more details of the task settings. The results in Table 2 demonstrate that PT generally achieves smaller average lengths compared to its base algorithm in most cases (9 out of 10) while maintaining the coverage. Besides, we conduct more experiments and ablations:

- We compare the length of RAPS and PT-RAPS under classification tasks in Table 4;

*Table 2.* Comparison of performance between VCP and PT-VCP in regression tasks across different datasets ($\alpha = 0.1$).

| METHOD | | VCP | | PT-VCP | |
| --- | --- | --- | --- | --- | --- |
| DATASET | BIAS | COVERAGE | LENGTH | COVERAGE | LENGTH |
| MEPS-19 | 20 | $0.90 \pm 0.000$ | $42.34 \pm 0.228$ | $0.90 \pm 0.000$ | $\mathbf{41.92} \pm 0.389$ |
| MEPS-20 | 20 | $0.90 \pm 0.000$ | $41.98 \pm 0.116$ | $0.90 \pm 0.000$ | $\mathbf{41.41} \pm 0.241$ |
| MEPS-21 | 20 | $0.90 \pm 0.004$ | $42.28 \pm 0.112$ | $0.90 \pm 0.000$ | $\mathbf{41.90} \pm 0.300$ |
| BIKE | 10 | $0.90 \pm 0.000$ | $20.46 \pm 0.018$ | $0.90 \pm 0.004$ | $\mathbf{19.59} \pm 0.018$ |
| BLOG-DATA | 20 | $0.90 \pm 0.004$ | $41.67 \pm 0.336$ | $0.90 \pm 0.000$ | $\mathbf{41.13} \pm 0.416$ |
| BIO | 10 | $0.90 \pm 0.004$ | $21.13 \pm 0.336$ | $0.90 \pm 0.000$ | $\mathbf{20.44} \pm 0.031$ |
| FACEBOOK-1 | 10 | $0.90 \pm 0.000$ | $20.81 \pm 0.036$ | $0.90 \pm 0.000$ | $\mathbf{20.80} \pm 0.179$ |
| FACEBOOK-2 | 10 | $0.90 \pm 0.000$ | $\mathbf{20.97} \pm 0.067$ | $0.90 \pm 0.000$ | $21.01 \pm 0.179$ |
| CONCRETE | 5 | $0.90 \pm 0.013$ | $10.32 \pm 0.009$ | $0.89 \pm 0.009$ | $\mathbf{9.87} \pm 0.031$ |
| STAR | 5 | $0.91 \pm 0.004$ | $10.14 \pm 0.004$ | $0.91 \pm 0.004$ | $\mathbf{9.63} \pm 0.027$ |

- We use CQR (Romano et al., 2019) as the base algorithm on regression tasks in Table 5;
- We evaluate a relaxed PT variant that replaces exactly null prediction sets with small nonempty prediction sets in Table 6;
- We evaluate PT-VCP under a larger bias setting in Table 7, where the interval-length contrast becomes more pronounced under stronger misspecification;
- We conduct ablation studies on hyperparameter $p$ and misspecification level $\mu$ in Appendix C.2.

### 3.5. Deceptive Improvement

We prove that PT preserves (conditional) coverage guarantees in Section 3.3 and achieves shorter prediction intervals under certain conditions in Section 3.4. Despite these theoretical benefits, PT is poorly suited for practical deployment. The primary issue is that PT introduces randomness, causing prediction intervals to vary across different runs. This inherent instability undermines the method's reliability. Besides, the problem becomes more dramatic in the scenario of Remark 3.1, where the individuals are grouped with different miscoverage rates. This randomness makes it impossible for a user to identify their assigned group, making the confidence interval meaningless. Therefore, **while PT may appear superior based on the traditional coverage-length metric, its practical instability makes it unsuitable for real-world deployment.** This discrepancy challenges the sufficiency of the coverage-length metric itself, suggesting it is not a complete measure of a method's practical utility.

**Beyond coverage-length.** Although our main analysis focuses on average interval length, the same mixture mechanism can affect other expectation-based efficiency criteria. To examine whether the phenomenon is specific to average length, we additionally evaluate RAPS and PT-RAPS under the p-value based efficiency criteria proposed by (Vovk et al., 2016). Across the ten criteria considered in that framework, PT-RAPS appears favorable on seven of them. These

results suggest that the failure mode is not limited to average set size: whenever an efficiency criterion averages a per-instance functional and does not sufficiently penalize degenerate or low-information prediction sets, the PT mixture can improve the reported value without improving per-instance reliability. The full results are reported in Table 9 and Table 10 in Appendix C.1.

## 4. Interval Stability

In this section, we propose *interval stability* (Definition 4.1) which measures the randomness in each run of CP. We begin with the definition of interval stability.

**Definition 4.1** (Interval Stability). Let $X$ denote a data point with returned confidence interval $C_{1-\alpha}(X)$, and let $|\cdot|$ denote a certain measure of the interval (e.g., its length). Let $\mathcal{A}$ denote the CP algorithm, and $\mathcal{D}_{ca}$ the calibration dataset. The interval stability is defined as

$$\mathrm{IS}(\mathcal{C}_{1-\alpha}(X)) \triangleq \mathbb{E}_X \left[ \mathrm{Var}_{\mathcal{A}|X,\mathcal{D}_{ca}}(\,|\mathcal{C}_{1-\alpha}(X)|\,) \right]. \quad (17)$$

The interval stability captures the expected variability of the interval size conditional on the test point and calibration randomness. Intuitively, it captures the inconsistency of the returned intervals when the algorithm is run multiple times on the same test point and calibration dataset.

Due to the stochastic nature of PT, it tends to produce a large interval stability, implying the practical instability issues. We prove in Proposition 4.2 that PT indeed introduces a non-zero interval stability.

**Proposition 4.2.** *Following the notations in Section 3.4, it holds that the interval stability is larger than zero:*

$$IS(\mathcal{C}_{1-\alpha}^{PT}(X)) = p(1-p)\left(\mathbb{E}\left(\mathcal{L}\left(\boldsymbol{x}, \frac{1-\tilde{\alpha}}{p}; s\right)^2\right)\right) > 0. \quad (18)$$

Notably, the *interval stability* metric is not just for detecting our specific PT construction, but serves as a safeguard to

*Table 3.* Interval stability comparison among VCP, PT-VCP, and localized CP on regression datasets.

| Method | meps-19 | meps-20 | meps-21 | bike | blog-data | bio | facebook-1 | facebook-2 | concrete | star |
|---|---|---|---|---|---|---|---|---|---|---|
| VCP | $0.00_{\pm0.000}$ | $0.00_{\pm0.000}$ | $0.00_{\pm0.000}$ | $0.00_{\pm0.000}$ | $0.00_{\pm0.000}$ | $0.00_{\pm0.000}$ | $0.00_{\pm0.000}$ | $0.00_{\pm0.000}$ | $0.00_{\pm0.000}$ | $0.00_{\pm0.000}$ |
| PT-VCP | $1.26_{\pm0.015}$ | $1.24_{\pm0.008}$ | $1.26_{\pm0.011}$ | $0.58_{\pm0.002}$ | $1.23_{\pm0.014}$ | $0.61_{\pm0.001}$ | $0.62_{\pm0.005}$ | $1.19_{\pm0.006}$ | $1.14_{\pm0.012}$ | $1.14_{\pm0.007}$ |
| Localized-CP | $0.63_{\pm0.336}$ | $0.34_{\pm0.273}$ | $0.90_{\pm0.594}$ | $0.03_{\pm0.012}$ | $1.41_{\pm1.083}$ | $0.03_{\pm0.014}$ | $0.64_{\pm0.579}$ | $0.40_{\pm0.266}$ | $0.11_{\pm0.076}$ | $0.01_{\pm0.007}$ |

ensure that future advancements in CP are genuine and reliable, rather than arising from the unprincipled randomness. As the community pushes for shorter prediction intervals, there is a risk that increasingly complex methods might implicitly introduce randomness that offers deceptive gains. See Remark 4.3 for detailed discussions.

*Remark* 4.3 (Why Interval Stability). Interval stability is still meaningful even if existing approaches in CP do not always rely on randomness. In the existing literature, numerous approaches claim a superior performance through a smaller interval length, under the traditional coverage-length metric. In this paper, we show that randomness may break this metric through PT due to practical issues. This raises concerns that as methods become increasingly complex, they may implicitly utilize similar randomness to improve the length. Such effects may be unintended but hard to recognize. To address this issue, interval stability serves as a complementary tool for detecting such issues and highlighting the risks inherent in the current reliance on the coverage-length metric alone.

**Experiment.** We evaluate Interval Stability on regression datasets in Table 3. The table compares three representative pipelines: VCP, PT-VCP, and localized CP. VCP serves as a deterministic baseline and has zero Interval Stability. PT-VCP introduces inference-time randomization and therefore exhibits substantially larger instability, showing that Interval Stability detects the vacuous randomness induced by PT. Localized CP, where the local scale estimator is retrained across runs, also exhibits nonzero run-to-run variation. This connects Interval Stability to a practical source of stochasticity, while not implying that the length advantage of localized CP is caused by PT-like behavior. Additional results for CQR and RAPS are provided in Table 11 and Table 12 in Appendix C.1.

Notably, interval stability is zero for deterministic methods by design. The metric is not intended to replace coverage and length, but to complement them, acting as a specific check against the kind of vacuous randomness we identify. A value of zero is a *pass* on this specific test, confirming the method's deterministic nature for a given input.

## 5. Conclusion

This paper demonstrates a potential risk of the coverage-length metric in CP. We introduce PT, a technique that hacks the conventional metric by producing deceptively shorter intervals while preserving coverage guarantees. However, PT relies on the randomness that leads to instability: the algorithm can produce different prediction sets for a given input on different runs. This creates practical issues in high-stakes scenarios. Our theoretical and empirical results confirm that while PT appears superior, its foundation is flawed. This discrepancy challenges the completeness of the coverage-length metric. Consequently, we propose Interval Stability as a complementary diagnostic tool, which helps flag the potential vacuous randomness for a newly proposed method.

## Acknowledgements

This work is supported by Shanghai Science and Technology Development Funds 24YF2711700 and Fundamental Research Funds for the Central Universities 2024110586.

## Impact Statement

This paper studies a failure mode in the evaluation of conformal prediction methods. The proposed Prejudicial Trick (PT) is not intended for deployment. Instead, it is introduced as a cautionary construction showing that coverage and average length can be manipulated by randomized prediction-set construction.

A direct deployment of PT or PT-like mechanisms could have negative societal consequences, especially in high-stakes domains such as healthcare, finance, education, or public services. Although marginal coverage may remain valid on average, some individuals may receive uninformative null or nearly null prediction sets due purely to algorithmic randomness. Such behavior can reduce reliability at the individual level and may create unfair treatment across users, even when each user has the same probability of being affected.

The broader goal of this work is therefore defensive: to encourage more careful evaluation of uncertainty quantification methods. We recommend that randomized conformal prediction pipelines report stability-related diagnostics, together with coverage, length, group or conditional coverage when relevant, and task-specific utility. Any method that improves average length by assigning uninformative prediction sets to a subset of samples should be avoided in practical deployment.

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

# Appendix

We firstly restate our contributions and demonstrate some additional related works, extended discussions, omitted preliminary and illustrations in Appendix A. Then we provide missing proofs in Appendix B. In Appendix C, we illustrate the omitted experimental results. In Appendix D, we present implementation details of our experiments.

## A. Additional Details and Discussion

### A.1. Contributions Restatement

We summarize our contributions as follows:

- We observe that the traditional coverage-length criteria in conformal prediction might be hacked using a counter-intuitive method PT, (Algorithm 1), since PT might deceptively improve the length while maintaining the coverage but raises fairness issues;

- We theoretically derive in Lemma 3.6 the conditions under which PT deceptively improves length, while keeping valid marginal coverage and conditional coverage (Theorem 3.3). We further derive several sufficient conditions under which PT improves length with first-order differentiability assumption (Theorem 3.7) or without first-order differentiability assumption (Theorem 3.10);

- We propose a new metric in Section 4, termed interval stability. Interval stability measures the variance of the prediction interval over the input introduced by the conformal prediction algorithms, helping to mitigate the adverse impacts of PT.

### A.2. Additional Related Works

**Conformal prediction.** Conformal prediction is a post hoc calibration framework that constructs statistically rigorous uncertainty sets for predictions from machine learning models (Vovk et al., 2005; Shafer & Vovk, 2008; Lei et al., 2018; Foygel Barber et al., 2021; Angelopoulos & Bates, 2021; Papadopoulos et al., 2008). Traditionally, vanilla conformal prediction is deployed in regression tasks (Vovk et al., 2005; Shafer & Vovk, 2008; Lei et al., 2018). Later, a branch of research expands vanilla conformal prediction to diverse data structures and applications, including classification tasks (Angelopoulos et al., 2020; Dabah & Tirer, 2025), censored data in survival analysis (Teng et al., 2021; Candès et al., 2023), functional data (Lei et al., 2015; Ajroldi et al., 2023), graph-based models (Zargarbashi et al., 2023; H Zargarbashi & Bojchevski, 2024), time series data (Xu & Xie, 2021; Stankeviciute et al., 2021), treatment effects (Lei & Candès, 2021; Jin et al., 2023), *etc*.

**Interval regression.** While coverage guarantees and interval length serve as fundamental metrics for evaluating conformal prediction (Vovk et al., 2005; Lei et al., 2018; Foygel Barber et al., 2021), these criteria are deeply entrenched in the broader paradigm of interval regression methodologies. Established approaches including quantile regression (Alaa et al., 2023; Sasaki et al., 2022) and Bayesian credible intervals (Kuleshov et al., 2018; Wang & Ghosal, 2023) similarly prioritize the dual metrics. Of particular relevance is Navratil et al. (2020) who proposes an excess and deficit metrics beyond the traditional coverage-length metric. Our paper differs from Navratil et al. (2020) in that our main contributions center on uncovering the inherent limitations of coverage-length metrics. Additionally, we contend that the proposed excess and deficit metrics cannot be directly applied to PT-VCP.

### A.3. More Discussions

**Similarity between PT and method discussed in Foygel Barber et al. (2021).** In Section 3.2, we mention the similarity between our PT method and the randomness discussed in Foygel Barber et al. (2021). While the mechanism in our work bears a structural resemblance to that in Foygel Barber et al. (2021), our motivation and conclusion are fundamentally different. Foygel Barber et al. (2021) investigate the inherent trade-offs required to achieve conditional coverage, using randomization as a tool to explore theoretical limits. In contrast, our work focuses on the evaluation paradigm itself. We use PT not to achieve a desirable property (like conditional coverage), but to demonstrate a failure mode of evaluating CP methods primarily through coverage and average length. Our primary contribution is to highlight this pitfall and to propose Interval Stability as a complementary diagnostic for the specific run-to-run variability induced by algorithmic randomness. This diagnostic perspective is orthogonal to the conditional-coverage tradeoffs studied by Foygel Barber et al. (2021).

## A.4. Omitted Preliminary

**Interval Prediction.** Interval prediction aims to construct a confidence interval that contains the true response value with a user-specified probability. Compared to traditional point estimation, interval prediction provides more comprehensive statistical information by quantifying the uncertainty using the interval length, which is often a more challenging goal. Definition A.1 presents the formal definition.

**Definition A.1** (Interval Prediction). Let $(X, Y)$ denote a feature-response pair. Given a miscoverage rate $\alpha$, interval prediction aims to construct a confidence interval $\mathcal{C}_{1-\alpha}(X)$, such that

$$\mathbb{P}(Y \in \mathcal{C}_{1-\alpha}(X)) \geq 1 - \alpha. \tag{19}$$

Given the coverage in Equation (19), a smaller confidence interval indicates a more precise estimate.

**Conformal Prediction.** To construct an interval prediction, we introduce a widely used approach called vanilla conformal prediction. The VCP method is typically divided into four stages: dataset splitting, training, calibration, and construction. The whole procedure is presented in Algorithm 2.

*Dataset Splitting.* Let $\mathcal{D} = \{(x_i, y_i) : i \in \mathcal{I}\}$ denote the i.i.d. samples from a distribution $\mathcal{P}_{XY}$ over the covariate $X \in \mathbb{R}^d$ and the response $Y \in \mathbb{R}$. The VCP first randomly splits the dataset $\mathcal{D}$ into two folds: a training fold $\mathcal{D}_{\text{tr}} = \{(x_i, y_i) : i \in \mathcal{I}_{\text{tr}}\}$ and a calibration fold $\mathcal{D}_{\text{ca}} = \{(x_i, y_i) : i \in \mathcal{I}_{\text{ca}}\}$, where $\mathcal{I}_{\text{tr}} \cup \mathcal{I}_{\text{ca}} = \mathcal{I}$ and $\mathcal{I}_{\text{tr}} \cap \mathcal{I}_{ca} = \varnothing$.

*Training Process.* We train a model denoted by $\hat{\mu}(\cdot)$ (*e.g.*, a neural network) via the training fold $\mathcal{D}_{\text{tr}}$.

*Calibration Process.* Given the trained model $\hat{\mu}(\cdot)$, VCP calculates the non-conformity score on the calibration fold $\mathcal{D}_{\text{ca}}$, denoted by $\mathcal{V} = \{s(x_i, y_i; \hat{\mu}) : i \in \mathcal{I}_{\text{ca}}\}$. The non-conformity score $s(\cdot)$ measures how well the model $\hat{\mu}(\cdot)$ fits the ground truth. A commonly used non-conformity score in regression tasks is the absolute residual, defined as $s(x_i, y_i; \hat{\mu}) = |y_i - \hat{\mu}(x_i)|$.

*Construction Process.* Finally, for a given miscoverage rate $\alpha$, we then compute a $(1 - \tilde{\alpha})$-th quantile $\hat{Q}_{1-\tilde{\alpha}}(\mathcal{V})$ of the empirical distribution of the non-conformity score set $\mathcal{V}$ calculated on the calibration set, where $1 - \tilde{\alpha} = (1 - \alpha)(1 + 1/|\mathcal{V}|)$. The prediction interval at a new point $x'$ is then given by

$$\mathcal{C}_{1-\alpha}(x') = \{y : s(x', y; \hat{\mu}) \leq \hat{Q}_{1-\tilde{\alpha}}(\mathcal{V})\}. \tag{20}$$

**Coverage and Length.** To evaluate the performance of interval prediction, two commonly used metrics: *coverage* and *length* are defined in Definition A.2, as further illustrated in Figure 2.

**Definition A.2** (Coverage and Length). Let $(X, Y)$ denote a feature-response pair from a joint distribution $\mathcal{P}_{XY}$, and let $\mathcal{C}_{1-\alpha}(X)$ denote the confidence interval to be evaluated and let $|\cdot|$ denote a certain measure of $\mathcal{C}_{1-\alpha}(X)$. The coverage and length of $\mathcal{C}_{1-\alpha}(X)$ is given by:

$$\begin{aligned} \text{Coverage} &:= \mathbb{E}\left[\mathbb{I}(Y \in \mathcal{C}_{1-\alpha}(X))\right], \\ \text{Length} &:= \mathbb{E}\left|\mathcal{C}_{1-\alpha}(X)\right|. \end{aligned} \tag{21}$$

For example, the length of the prediction interval given by VCP in Equation (20) is:

$$\text{Length} = \mathbb{E}\left[2\hat{Q}_{1-\tilde{\alpha}}(\mathcal{V})\right]. \tag{22}$$

Notably, the two metrics in Definition A.2 evaluate the quality of prediction intervals from different perspectives. Figure 2 illustrates the coverage and length given a distribution. Firstly, high coverage ensures that the true value falls within the interval with high probability. A valid confidence interval should guarantee that the coverage exceeds $1 - \alpha$, as suggested in Equation 19. However, setting a sufficiently large interval always guarantees Equation (19), which is impractical and meaningless. Therefore, the length metric is required to ensure the interval's precision. Based on the above discussion, the gold standard in conformal prediction is *making the length as small as possible, given that the coverage is larger than* $1 - \alpha$.

Following the gold standard, VCP ensures the coverage guarantee under mild exchangeability assumption (Proposition A.3), but pays less attention to the length. As a result, numerous works on improving the length of VCP from different perspectives (Papadopoulos et al., 2011; Romano et al., 2019) use intuitively valid approaches.

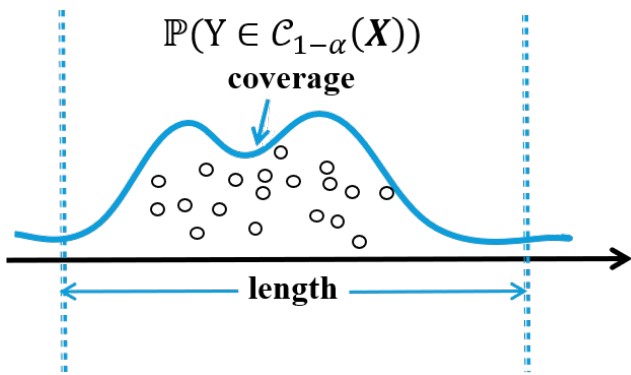

*Figure 2.* Illustration of coverage and interval length.

**Proposition A.3** (Coverage Guarantee). *The terms $\mathcal{U}_i$ are exchangeable if arbitrary permutation leads to the same distribution, i.e., $(\mathcal{U}_1, ..., \mathcal{U}_{|\mathcal{I}_{ca}|+1}) \overset{d}{=} (\mathcal{U}_{\pi(1)}, ..., \mathcal{U}_{\pi(|\mathcal{I}_{ca}|+1)})$ with arbitrary permutation $\pi$ over $1, ..., |\mathcal{I}_{ca} + 1|$, where $\overset{d}{=}$ denotes equivalence in distribution. Suppose that the data pair $(\boldsymbol{x}_i, y_i), i \in \mathcal{I}_{ca}$ and the test point $(\boldsymbol{x}', y')$ are exchangeable, then the confidence interval $\mathcal{C}_{1-\alpha}(\boldsymbol{x}')$ returned by Algorithm 2 satisfies*

$$\mathbb{P}\left(y' \in \mathcal{C}_{1-\alpha}(\boldsymbol{x}')\right) \geq 1 - \alpha.$$

### A.5. Missing Illustration

In this section, we present the missing illustration of Example 1 in Section 1, the illustration of PT (Figure 4) and VCP algorithm mentioned in Section 3.1.

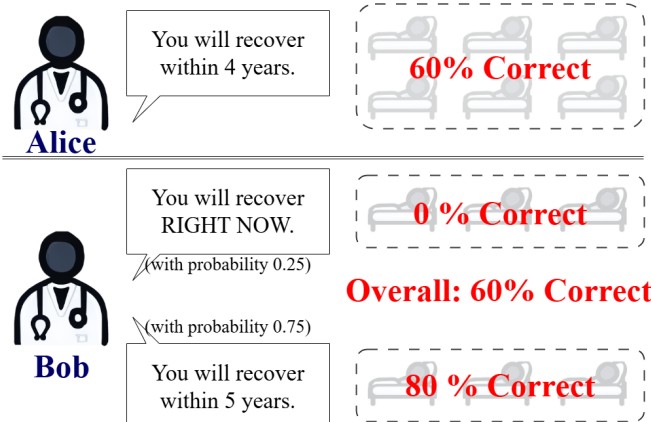

*Figure 3.* Illustration of Example 1. Doctor Alice and Bob both achieve 60% accuracy. Bob is more precise regarding length, but the corresponding strategy is not practically valid.

---

**Algorithm 2** Vanilla Conformal Prediction (VCP)

---

1: **Input:** miscoverage rate $\alpha$, dataset $\mathcal{D} = \{(\boldsymbol{x}_i, y_i) : i \in \mathcal{I}\}$, test point $\boldsymbol{x}'$, non-conformity score function $s(\boldsymbol{x}_i, y_i; \hat{\mu})$.
2: Randomly split $\mathcal{D}$ into a training fold $\mathcal{D}_{tr} = \{(\boldsymbol{x}_i, y_i) : i \in \mathcal{I}_{tr}\}$ and a calibration fold $\mathcal{D}_{ca} = \{(\boldsymbol{x}_i, y_i) : i \in \mathcal{I}_{ca}\}$;
3: Train a model $\hat{\mu}$ based on the training fold $\mathcal{D}_{tr}$;
4: Calculate the non-conformity score on the calibration fold $\mathcal{D}_{ca}$, denoted by $\mathcal{V} = \{s(\boldsymbol{x}_i, y_i; \hat{\mu}) : i \in \mathcal{I}_{ca}\}$;
5: Compute the $(1 - \tilde{\alpha})$-th quantile $\hat{Q}_{1-\tilde{\alpha}}(\mathcal{V})$ of the empirical distribution of the non-conformity score set $\mathcal{V}$ calculated on the calibration set $\mathcal{D}_{ca}$, where $1 - \tilde{\alpha} = (1 - \alpha)(1 + 1/|\mathcal{V}|)$;
6: **Output:** Interval $\mathcal{C}_{1-\alpha}(\boldsymbol{x}') = \{y : s(\boldsymbol{x}', y; \hat{\mu}) \leq \hat{Q}_{1-\tilde{\alpha}}(\mathcal{V})\}$.

---

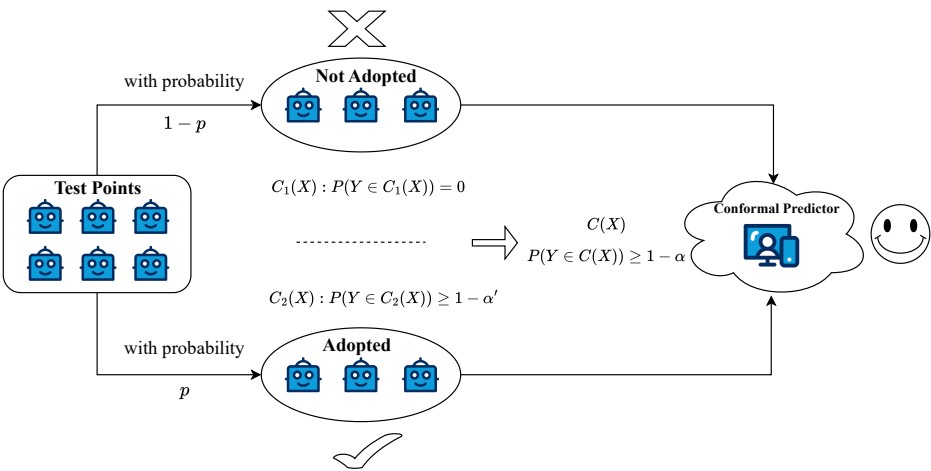

*Figure 4.* The illustration of Prejudicial Trick (PT). To obtain a $1 - \alpha$ confidence interval, PT first assigns empty sets for a $1 - p$ subset of the test points, and assigns $1 - \alpha'$ confidence interval for the remaining test points where $\alpha' < \alpha$. The returned confidence interval still satisfies $\mathbb{P}(Y \in \mathcal{C}(X)) \geq 1 - \alpha$ by setting a proper $\alpha'$.

# B. Proofs for Theorems and Corollaries

### B.1. Proof of Theorem 3.3

*Proof.* We first prove the marginal coverage guarantee. By the construction of PT in Algorithm 1, the returned interval is the null set with probability $1 - p$, and is the base interval $C_{1-\alpha'}^{\mathrm{CP}}(X')$ with probability $p$, where

$$\alpha' = 1 - \frac{1 - \alpha}{p}. \tag{23}$$

Therefore,

$$\mathbb{P}\left(y' \in C_{1-\alpha}^{\mathrm{PT}}(X')\right) \geq p \cdot \mathbb{P}\left(y' \in C_{1-\alpha'}^{\mathrm{CP}}(X')\right). \tag{24}$$

Under the exchangeability assumption, the base CP algorithm satisfies the marginal coverage guarantee at level $1 - \alpha'$, namely

$$\mathbb{P}\left(y' \in C_{1-\alpha'}^{\mathrm{CP}}(X')\right) \geq 1 - \alpha'. \tag{25}$$

Hence,

$$\begin{aligned} \mathbb{P}\left(y' \in C_{1-\alpha}^{\mathrm{PT}}(X')\right) &\geq p(1 - \alpha') \\ &= p \cdot \frac{1 - \alpha}{p} \\ &= 1 - \alpha. \end{aligned} \tag{26}$$

This proves the marginal coverage guarantee.

We next prove the conditional coverage guarantee. Suppose that the base algorithm satisfies conditional coverage for any miscoverage level. Applying this assumption at level $\alpha'$, we have

$$\mathbb{P}\left(y' \in C_{1-\alpha'}^{\mathrm{CP}}(X') \mid X'\right) \geq 1 - \alpha' \tag{27}$$

for $X'$ almost surely. Since the randomization in Algorithm 1 is independent of the test point and the response, conditioning on $X'$ gives

$$\begin{aligned} \mathbb{P}\left(y' \in C_{1-\alpha}^{\mathrm{PT}}(X') \mid X'\right) &\geq p \cdot \mathbb{P}\left(y' \in C_{1-\alpha'}^{\mathrm{CP}}(X') \mid X'\right) \\ &\geq p(1 - \alpha') \\ &= 1 - \alpha. \end{aligned} \tag{28}$$

Therefore,

$$\mathbb{P}\left(y' \in C_{1-\alpha}^{\mathrm{PT}}(X') \mid X'\right) \geq 1 - \alpha \tag{29}$$

holds for any $\alpha$ and for $X'$ almost surely. This completes the proof. □

### B.2. Proof of Lemma 3.6

*Proof.* Let $q = 1 - \tilde{\alpha}$ denote the target coverage level. By Algorithm 1, PT returns the null set with probability $1 - p$, whose measure is zero, and returns the base CP interval at the adjusted coverage level $q/p$ with probability $p$. Therefore,

$$
\begin{aligned}
\mathbb{E}\left[|C_{1-\tilde{\alpha}}^{\mathrm{PT}}(X')|\right] &= (1-p) \cdot 0 + p\,\mathbb{E}_{X'}\left[L\left(X', \frac{q}{p}; s\right)\right] \\
&= p\,\mathcal{G}\left(\frac{q}{p}; s\right) \\
&= p\,\mathcal{G}\left(\frac{1-\tilde{\alpha}}{p}; s\right).
\end{aligned}
\tag{30}
$$

On the other hand, the expected length of the base CP algorithm at the target coverage level $q = 1 - \tilde{\alpha}$ is

$$
\begin{aligned}
\mathbb{E}\left[|C_{1-\tilde{\alpha}}^{\mathrm{CP}}(X')|\right] &= \mathbb{E}_{X'}\left[L\left(X', 1 - \tilde{\alpha}; s\right)\right] \\
&= \mathcal{G}(1 - \tilde{\alpha}; s).
\end{aligned}
\tag{31}
$$

Hence, if there exists $p \in (1 - \tilde{\alpha}, 1)$ such that

$$p\,\mathcal{G}\left(\frac{1-\tilde{\alpha}}{p}; s\right) < \mathcal{G}(1 - \tilde{\alpha}; s), \tag{32}$$

then

$$
\begin{aligned}
\mathbb{E}\left[|C_{1-\tilde{\alpha}}^{\mathrm{PT}}(X')|\right] &= p\,\mathcal{G}\left(\frac{1-\tilde{\alpha}}{p}; s\right) \\
&< \mathcal{G}(1 - \tilde{\alpha}; s) \\
&= \mathbb{E}\left[|C_{1-\tilde{\alpha}}^{\mathrm{CP}}(X')|\right].
\end{aligned}
\tag{33}
$$

This proves that PT has a shorter expected length than its base algorithm under the stated condition. □

### B.3. Proof of Theorem 3.7

*Proof.* Let $q = 1 - \tilde{\alpha}$. By Lemma 3.6, it suffices to show that there exists $p \in (q, 1)$ such that

$$p\,\mathcal{G}\left(\frac{q}{p}; s\right) < \mathcal{G}(q; s). \tag{34}$$

Equivalently, by writing $u = q/p$, it suffices to find some $u \in (q, 1)$ such that

$$\frac{\mathcal{G}(u; s)}{u} < \frac{\mathcal{G}(q; s)}{q}. \tag{35}$$

Define

$$H(u) := \frac{\mathcal{G}(u; s)}{u}. \tag{36}$$

Since $\mathcal{G}(\cdot; s)$ is first-order differentiable at $q$, $H$ is also differentiable at $q$, and

$$H'(q) = \frac{q\frac{\partial}{\partial u}\mathcal{G}(u; s)\big|_{u=q} - \mathcal{G}(q; s)}{q^2}. \tag{37}$$

By the assumption in Lemma 3.6, we have

$$\frac{\mathcal{G}(q;s)}{q} > \frac{\partial}{\partial u}\mathcal{G}(u;s)\Big|_{u=q}, \tag{38}$$

which implies

$$q\frac{\partial}{\partial u}\mathcal{G}(u;s)\Big|_{u=q} - \mathcal{G}(q;s) < 0. \tag{39}$$

Therefore,

$$H'(q) < 0. \tag{40}$$

Hence, there exists some $u \in (q, 1)$ sufficiently close to $q$ such that

$$H(u) < H(q), \tag{41}$$

or equivalently,

$$\frac{\mathcal{G}(u;s)}{u} < \frac{\mathcal{G}(q;s)}{q}. \tag{42}$$

Taking $p = q/u$, we have $p \in (q, 1)$ and $q/p = u$. Thus,

$$p\mathcal{G}\left(\frac{q}{p};s\right) = \frac{q}{u}\mathcal{G}(u;s)$$
$$< \mathcal{G}(q;s). \tag{43}$$

By Lemma 3.6, this implies

$$\mathbb{E}\left[\left|C_{1-\tilde{\alpha}}^{\mathrm{PT}}(X')\right|\right] < \mathbb{E}\left[\left|C_{1-\tilde{\alpha}}^{\mathrm{CP}}(X')\right|\right]. \tag{44}$$

This completes the proof. $\square$

### B.4. Proof of Theorem 3.10

*Proof.* Let

$$q = 1 - \tilde{\alpha}. \tag{45}$$

By the assumption in Theorem 3.10, there exists $u \in (q, 1)$ such that

$$\frac{\mathcal{G}(q;s)}{q} > \frac{\mathcal{G}(u;s) - \mathcal{G}(q;s)}{u - q}. \tag{46}$$

Since $u > q > 0$, multiplying both sides by $q(u - q)$ gives

$$(u - q)\mathcal{G}(q;s) > q\left(\mathcal{G}(u;s) - \mathcal{G}(q;s)\right). \tag{47}$$

Rearranging the terms yields

$$u\mathcal{G}(q;s) > q\mathcal{G}(u;s). \tag{48}$$

Equivalently,

$$\frac{q}{u}\mathcal{G}(u;s) < \mathcal{G}(q;s). \tag{49}$$

Now take

$$p = \frac{q}{u} = \frac{1 - \tilde{\alpha}}{u}. \tag{50}$$

Since $u \in (q, 1)$, we have $p \in (q, 1)$. Moreover,

$$\frac{q}{p} = u. \tag{51}$$

Therefore,

$$p\mathcal{G}\left(\frac{q}{p}; s\right) = \frac{q}{u}\mathcal{G}(u; s)$$
$$< \mathcal{G}(q; s). \tag{52}$$

By Lemma 3.6, this implies

$$\mathbb{E}\left[\left|C_{1-\tilde{\alpha}}^{\mathrm{PT}}(X')\right|\right] < \mathbb{E}\left[\left|C_{1-\tilde{\alpha}}^{\mathrm{CP}}(X')\right|\right]. \tag{53}$$

This completes the proof. $\square$

### B.5. Proof of Example 3

*Proof.* Let $q = 1 - \tilde{\alpha}$. For VCP with the absolute residual score, the prediction interval at coverage level $u$ has radius equal to the $u$-quantile of the non-conformity score $S$. By assumption,

$$\mathbb{P}(S \leq r) = r^2, \qquad r \in [0, 1]. \tag{54}$$

Hence, the population $u$-quantile $r_u$ of $S$ satisfies

$$u = \mathbb{P}(S \leq r_u)$$
$$= r_u^2. \tag{55}$$

Therefore,

$$r_u = \sqrt{u}. \tag{56}$$

Since the VCP interval has radius $r_u$, its length is $2r_u$. Thus, the averaged length function is

$$\mathcal{G}(u; s) = 2r_u$$
$$= 2\sqrt{u}. \tag{57}$$

For PT with parameter $p \in (q, 1)$, the adjusted coverage level is $q/p$. Therefore,

$$p\mathcal{G}\left(\frac{q}{p}; s\right) = p \cdot 2\sqrt{\frac{q}{p}}$$
$$= 2\sqrt{pq}. \tag{58}$$

Since $p < 1$, we have

$$2\sqrt{pq} < 2\sqrt{q}. \tag{59}$$

Moreover,

$$\mathcal{G}(q; s) = 2\sqrt{q}. \tag{60}$$

Combining the above equations gives

$$p\mathcal{G}\left(\frac{q}{p}; s\right) < \mathcal{G}(q; s). \tag{61}$$

By Lemma 3.6, PT strictly reduces the expected length compared with the base CP algorithm:

$$\mathbb{E}\left[\left|C_{1-\tilde{\alpha}}^{\mathrm{PT}}(X')\right|\right] < \mathbb{E}\left[\left|C_{1-\tilde{\alpha}}^{\mathrm{CP}}(X')\right|\right]. \tag{62}$$

This completes the proof. $\square$

## B.6. Proof of Example 4

When the non-conformity score follows a Gaussian distribution, the analytical solutions of the interval length returned by VCP and PT-VCP are

$$|\mathcal{C}_{1-\tilde{\alpha}}^{VCP}(\boldsymbol{X}')| = 2\Phi^{-1}\left(1 - \frac{\tilde{\alpha}}{2}\right), \quad |\mathcal{C}_{1-\tilde{\alpha}}^{PT}(\boldsymbol{X}')| = 2p\Phi^{-1}\left(1 - \frac{1}{2}\left(1 - \frac{1-\tilde{\alpha}}{p}\right)\right) \tag{63}$$

where $\Phi(\cdot)$ is the cumulative distribution function of the Gaussian distribution. Therefore, PT fails since Lemma B.1.

**Lemma B.1.** $\forall \alpha \in (0,1), p \in (1-\alpha, 1)$, there holds

$$\Phi^{-1}\left(1 - \frac{\tilde{\alpha}}{2}\right) < p\Phi^{-1}\left(1 - \frac{1}{2}\left(1 - \frac{1-\tilde{\alpha}}{p}\right)\right) \tag{64}$$

*Proof.* Using the symmetry identity $\Phi^{-1}(1-u) = -\Phi^{-1}(u)$, the desired inequality is equivalent to

$$p\,\Phi^{-1}\left(\tfrac{1}{2}\left(1 - \tfrac{1-\alpha}{p}\right)\right) < \Phi^{-1}(\alpha/2). \tag{65}$$

Define

$$u_0 := \alpha/2 \in (0,1/2), \qquad u(p) := \tfrac{1}{2}\left(1 - \tfrac{1-\alpha}{p}\right) \in (0,1/2). \tag{66}$$

Let $g(u) := \Phi^{-1}(u)$ on $(0,1/2)$. Since

$$g'(u) = \frac{1}{\phi(\Phi^{-1}(u))} > 0, \qquad g''(u) = \frac{\Phi^{-1}(u)}{\phi(\Phi^{-1}(u))^2} < 0, \tag{67}$$

where $\phi(\cdot)$ denotes the p.d.f. of the standard normal distribution, the function $g$ is increasing and strictly concave. Hence, for any $u \leq u_0$,

$$g(u) \leq g(u_0) + g'(u_0)(u - u_0). \tag{68}$$

Plugging $u = u(p)$ into Eq (68) and multiplying both sides by $p \in (0,1)$ gives

$$p\,g(u(p)) \leq p\,g(u_0) + p\,g'(u_0)\left(u(p) - u_0\right). \tag{69}$$

A direct calculation yields

$$u(p) - u_0 = \frac{(1-\alpha)(p-1)}{2p} < 0. \tag{70}$$

Subtracting $g(u_0)$ from both sides of Eq (69) and using Eq (70), we obtain

$$p\,g(u(p)) - g(u_0) \leq (1-p)\left[-g(u_0) + \frac{1-\alpha}{2\,\phi(g(u_0))}\right]. \tag{71}$$

Here $1 - p > 0$. Moreover, since $g(u_0) = \Phi^{-1}(\alpha/2) < 0$ and $\phi(g(u_0)) > 0$, the bracket in Eq (71) is nonnegative. Thus the right-hand side of Eq (71) is nonpositive, implying

$$p\,g(u(p)) - g(u_0) < 0, \tag{72}$$

which is exactly inequality Eq (65). The inequality is strict because $u(p) \neq u_0$ and $g$ is strictly concave.

Reverting to the upper-tail form via $\Phi^{-1}(1-u) = -\Phi^{-1}(u)$ completes the proof:

$$\Phi^{-1}(1 - \alpha/2) < p\,\Phi^{-1}\left(\tfrac{1}{2}\left(1 + \tfrac{1-\alpha}{p}\right)\right). \tag{73}$$

$\square$

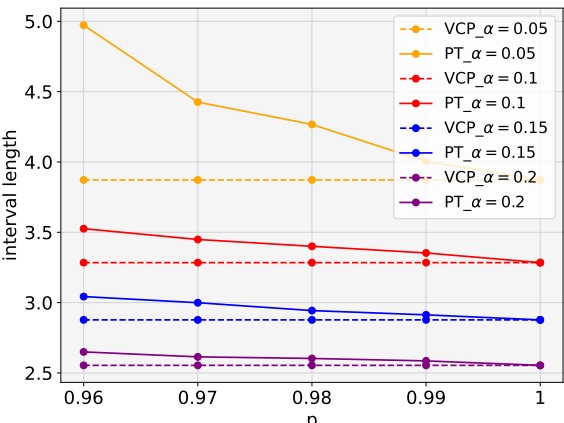

*Figure 5.* PT fails to improve length when the conditions on the distribution of non-conformity score are not satisfied.

*Table 4.* Comparison between RAPS and PT-RAPS in classification tasks across different models, with $\alpha = 0.1, p = 0.95$ and index range chosen as 300.

| METHOD | BIAS | RAPS | | PT-RAPS | |
|---|---|---|---|---|---|
| MODEL | | COVERAGE | LENGTH | COVERAGE | LENGTH |
| RESNET18 | 40 | 0.90 ±0.000 | 304.02 ±0.004 | 0.90 ±0.000 | **295.60** ±0.233 |
| RESNET50 | 40 | 0.90 ±0.000 | 302.09 ±0.027 | 0.90 ±0.000 | **290.29** ±0.224 |
| RESNET101 | 40 | 0.90 ±0.000 | 302.01 ±0.004 | 0.90 ±0.000 | **289.56** ±0.174 |
| RESNET152 | 40 | 0.89 ±0.004 | 301.53 ±0.054 | 0.90 ±0.000 | **288.98** ±0.165 |
| RESNEXT101 | 40 | 0.90 ±0.000 | 301.48 ±0.013 | 0.90 ±0.000 | **288.49** ±0.201 |
| VGG16 | 40 | 0.90 ±0.004 | 303.39 ±0.143 | 0.90 ±0.000 | **293.34** ±0.304 |
| SHUFFLENET | 40 | 0.90 ±0.000 | 304.05 ±0.040 | 0.90 ±0.000 | **295.79** ±0.282 |
| INCEPTION | 40 | 0.90 ±0.000 | 304.10 ±0.013 | 0.90 ±0.000 | **297.25** ±0.228 |
| DENSENET161 | 40 | 0.90 ±0.000 | 302.03 ±0.009 | 0.90 ±0.000 | **289.29** ±0.197 |

# C. Omitted Experiments

In this section, we present all the omitted experiments. In Appendix C.1, we demonstrate the missing experimental results in Section 3.3 and Section 3.4. In Appendix C.2, we exhibit the ablation study results.

### C.1. Omitted Experimental Results

**Classification Tasks.** We extend PT to classification tasks. We apply PT to the real-world IMAGENET-VAL dataset (Deng et al., 2009) with several pre-trained models, a similar setting with Angelopoulos et al. (2020). To simulate model misspecification, a bias is introduced to the logits of several classes before the softmax operation. The magnitude of the bias is determined based on the scale of the outputs. The experimental results on classification tasks in Table 4 perform similarly to regression tasks. Specifically, PT-RAPS attains valid coverage across different models (Theorem 3.3) while reducing the set size compared to RAPS (Theorem 3.10).

**Conformalized Quantile Regression.** We deploy PT into other variants of conformal prediction. Specifically, we choose CQR as a baseline (Romano et al., 2019). CQR inherits the advantages of both conformal prediction and classical quantile regression. We use the same datasets and evaluation metrics as in Section 3.4. To mimic the model misspecification, we add bias directly to the lower and upper quantiles obtained by the quantile regression. The experimental results on the real-world CQR tasks are exhibited in Table 5. It illustrates that *PT achieves shorter interval length while maintaining valid coverage on CQR.*

**Relaxed PT without exactly null prediction sets.** The standard PT construction assigns a null prediction set to a fraction of test samples. To examine whether the phenomenon relies on exactly null outputs, we further consider a relaxed PT

*Table 5.* Comparison between CQR and PT-CQR in quantile regression task across different datasets.

| Method | Bias | CQR | | PT-CQR | |
|---|---|---|---|---|---|
| Dataset | | Coverage | Length | Coverage | Length |
| meps-19 | 1 | 0.91 ±0.000 | 4.60 ±0.148 | 0.91 ±0.246 | **4.44** ±0.143 |
| meps-20 | 1 | 0.91 ±0.000 | 4.58 ±0.192 | 0.91 ±0.179 | **4.41** ±0.188 |
| meps-21 | 1 | 0.91 ±0.000 | 4.65 ±0.080 | 0.91 ±0.161 | **4.52** ±0.107 |
| bike | 1 | 0.91 ±0.000 | 2.61 ±0.013 | 0.90 ±0.268 | **2.51** ±0.009 |
| blog-data | 1 | 0.91 ±0.000 | 3.80 ±0.107 | 0.93 ±0.116 | **3.61** ±0.098 |
| bio | 1 | 0.91 ±0.000 | 3.45 ±0.009 | 0.90 ±0.112 | **3.32** ±0.009 |
| facebook-1 | 1 | 0.91 ±0.000 | 3.38 ±0.022 | 0.92 ±0.125 | **3.22** ±0.027 |
| facebook-2 | 1 | 0.91 ±0.000 | 3.57 ±0.027 | 0.92 ±0.085 | **3.39** ±0.027 |
| concrete | 2 | 0.91 ±0.000 | 4.39 ±0.022 | 0.88 ±0.648 | **4.23** ±0.018 |
| star | 2 | 0.91 ±0.000 | 4.15 ±0.004 | 0.90 ±0.349 | **3.96** ±0.009 |

*Table 6.* Comparison of performance between VCP and localized-CP-like PT-VCP in regression tasks across different datasets at fixed $\alpha = 0.1$, $p = 0.96$, and perturbation rate $\epsilon = 0.01$. Values are reported as mean $\pm$ std over 5 random seeds. Bold marks the smaller interval length between VCP and PT-VCP for each dataset.

| Method | Bias | VCP | | PT-VCP | |
|---|---|---|---|---|---|
| Dataset | | Coverage | Length | Coverage | Length |
| meps-19 | 20 | 0.901 ±0.005 | 42.34 ±0.51 | 0.896 ±0.003 | **41.73** ±0.68 |
| meps-20 | 20 | 0.900 ±0.005 | 41.98 ±0.26 | 0.901 ±0.006 | **41.40** ±0.42 |
| meps-21 | 20 | 0.903 ±0.005 | 42.28 ±0.25 | 0.897 ±0.003 | **41.62** ±0.48 |
| bike | 10 | 0.901 ±0.005 | 20.46 ±0.04 | 0.901 ±0.005 | **19.79** ±0.08 |
| blog-data | 20 | 0.898 ±0.005 | 41.67 ±0.75 | 0.899 ±0.003 | **41.04** ±0.86 |
| bio | 10 | 0.901 ±0.004 | 21.13 ±0.05 | 0.900 ±0.003 | **20.55** ±0.06 |
| facebook-1 | 10 | 0.902 ±0.003 | 20.81 ±0.08 | 0.899 ±0.003 | **20.62** ±0.24 |
| facebook-2 | 10 | 0.900 ±0.001 | 20.97 ±0.15 | 0.900 ±0.003 | **20.87** ±0.30 |
| concrete | 5 | 0.895 ±0.030 | 10.32 ±0.02 | 0.899 ±0.021 | **10.09** ±0.09 |
| star | 5 | 0.909 ±0.010 | 10.14 ±0.01 | 0.909 ±0.002 | **9.78** ±0.02 |

variant that replaces the null prediction set with a small nonempty prediction set controlled by a perturbation rate $\epsilon$. This construction is closer to localized CP in the sense that the returned set can be viewed as being generated by a very small but nonzero local scale estimate, rather than by an exactly zero scale. As shown in Table 6, the relaxed PT-VCP still preserves comparable coverage while reducing the average interval length across all datasets. This suggests that the observed metric failure is not merely an artifact of returning exactly null prediction sets.

**Stronger misspecification.** We also evaluate PT-VCP under a larger bias setting to further investigate how model misspecification affects the length reduction induced by PT. This experiment complements the main regression results in Section 3.4, where the improvement in average length can be modest on some datasets. As reported in Table 7, increasing the bias amplifies the interval-length contrast between VCP and PT-VCP while maintaining comparable coverage. These results are consistent with our theoretical discussion that PT is more likely to appear favorable when the sufficient conditions for length reduction are more strongly satisfied.

**Group Coverage.** In Section 3.3, we conduct experiments to evaluate the different performance of VCP and PT-VCP regarding group coverage. The experimental results shown in Table 8 demonstrate that PT not only achieves shorter confidence intervals while maintaining overall coverage, *but also improves the group coverage in regression tasks*[5].

**Additional p-value based efficiency criteria.** We further evaluate RAPS and PT-RAPS under the p-value based efficiency criteria proposed by (Vovk et al., 2016). These criteria provide alternative summaries of predictive efficiency for classification. As shown in Tables 9 and 10, PT-RAPS obtains more favorable values on seven out of the ten criteria. This supports the conclusion that the PT failure mode is not restricted to the average set-size metric. However, this experiment should be interpreted as empirical evidence rather than a complete characterization of all possible efficiency criteria.

**Interval Stability.** In Section 4, we introduce a new evaluation criterion, termed *interval stability* and conduct several

---

[5]Group coverage is defined as the lowest coverage rate among all the groups.

*Table 7.* Comparison of performance between VCP and PT-VCP in regression tasks across different datasets at fixed $\alpha = 0.1$ and $p = 0.96$, using a larger bias setting to amplify the interval-length contrast. Values are reported as mean $\pm$ std over 5 random seeds. Bold marks the smaller interval length between VCP and PT-VCP for each dataset.

| Method | Bias | VCP | | PT-VCP | |
| --- | --- | --- | --- | --- | --- |
| Dataset | | Coverage | Length | Coverage | Length |
| meps-19 | 80 | 0.900 $\pm$0.006 | 162.32 $\pm$0.51 | 0.898 $\pm$0.006 | **156.99** $\pm$0.65 |
| meps-20 | 80 | 0.900 $\pm$0.005 | 161.96 $\pm$0.26 | 0.901 $\pm$0.003 | **156.62** $\pm$0.65 |
| meps-21 | 80 | 0.904 $\pm$0.005 | 162.27 $\pm$0.25 | 0.899 $\pm$0.003 | **156.88** $\pm$0.75 |
| bike | 40 | 0.901 $\pm$0.005 | 80.46 $\pm$0.04 | 0.899 $\pm$0.006 | **77.29** $\pm$0.18 |
| blog-data | 80 | 0.899 $\pm$0.005 | 161.64 $\pm$0.73 | 0.899 $\pm$0.004 | **156.10** $\pm$0.92 |
| bio | 40 | 0.901 $\pm$0.004 | 81.13 $\pm$0.05 | 0.900 $\pm$0.003 | **78.11** $\pm$0.14 |
| facebook-1 | 40 | 0.902 $\pm$0.002 | 80.77 $\pm$0.07 | 0.901 $\pm$0.003 | **78.15** $\pm$0.28 |
| facebook-2 | 40 | 0.900 $\pm$0.001 | 80.94 $\pm$0.15 | 0.899 $\pm$0.001 | **78.33** $\pm$0.32 |
| concrete | 20 | 0.895 $\pm$0.030 | 40.32 $\pm$0.02 | 0.892 $\pm$0.021 | **38.65** $\pm$0.21 |
| star | 20 | 0.909 $\pm$0.010 | 40.14 $\pm$0.01 | 0.906 $\pm$0.009 | **38.47** $\pm$0.27 |

*Table 8.* Comparison of group coverage between VCP and PT-VCP on regression tasks across different datasets ($\alpha = 0.1$).

| Dataset | Group | VCP | PT-VCP |
| --- | --- | --- | --- |
| bike | Day | 0.878 $\pm$ 0.007 | **0.884** $\pm$ 0.010 |
| | Month | 0.826 $\pm$ 0.010 | **0.857** $\pm$ 0.011 |
| | Year | 0.851 $\pm$ 0.005 | **0.871** $\pm$ 0.004 |
| star | Gender | 0.905 $\pm$ 0.008 | **0.905** $\pm$ 0.002 |
| | Stark | 0.890 $\pm$ 0.005 | **0.895** $\pm$ 0.007 |
| | School1 | 0.902 $\pm$ 0.008 | **0.899** $\pm$ 0.022 |
| meps-19 | SEX=1 | 0.883 $\pm$ 0.004 | **0.895** $\pm$ 0.001 |
| | MARRY=1 | 0.901 $\pm$ 0.004 | **0.901** $\pm$ 0.003 |
| | REGION=1 | 0.862 $\pm$ 0.005 | **0.877** $\pm$ 0.006 |
| meps-20 | FTSTU=1 | 0.893 $\pm$ 0.004 | **0.900** $\pm$ 0.002 |
| | ACTDTY=1 | 0.897 $\pm$ 0.003 | **0.902** $\pm$ 0.002 |
| | HONRDC=1 | 0.792 $\pm$ 0.010 | **0.846** $\pm$ 0.008 |
| meps-21 | RTHLTH=1 | 0.864 $\pm$ 0.004 | **0.877** $\pm$ 0.004 |
| | MNHLTH=1 | 0.856 $\pm$ 0.004 | **0.873** $\pm$ 0.004 |
| | HIBPDX=1 | 0.755 $\pm$ 0.013 | **0.818** $\pm$ 0.009 |

empirical evaluations using the datasets described in Section 3.4, the results of which are listed in Table 3. We further investigate the performance of the interval stability metric using CQR as the base algorithm in Table 11. The results are similar to the results in Table 3. We also evaluate the interval stability metric on classification tasks. As shown in Table 12, interval stability successfully identifies the vacuous randomness in PT.

### C.2. Ablation Studies

This section exhibits the ablation studies on the probability hyperparameter $p$ in PT and the bias parameter $\mu$ on different base algorithms (Figure 6-Figure 15). All the experiments are conducted based on various miscoverage rates $\alpha$. The experiment results demonstrate that, although not all the probability hyperparameters $p$ outperform the base algorithm, our goal is to show that *there exist multiple (at least one) probability hyperparameters such that PT-VCP outperforms VCP, which suffices to challenge the coverage-length gold standard.* Furthermore, we find that the bias parameter actually matters here, implying that PT-VCP performs better than VCP under misspecification, which validates Theorem 3.10.

## D. Experiment Details

In this section, we provide implementation details of the experiments in this paper, including experiments on synthetic datasets in Appendix D.1 and experiments on real-world datasets in Appendix D.2.

*Table 9.* Main p-value criteria across models. Hyperparameters are fixed as $\alpha = \epsilon = 0.1$, calibration size $= 10{,}000$, seeds $= 5$, $p = 0.95$, `pt_bias` $= 40$, `pt_index_range` $= 300$, with smoothed p-values enabled. Each entry is reported as RAPS / PT-RAPS. Bold indicates the better, i.e., smaller, value between RAPS and PT-RAPS for each metric/model pair.

| Model | S | N | U | F | M | E |
|---|---|---|---|---|---|---|
| ResNet18 | **215.01** ±1.12 / 241.49 ±0.89 | 304.02 ±0.01 / **295.44** ±0.20 | 0.93 ±0.00 / **0.89** ±0.00 | **214.05** ±1.12 / 240.59 ±0.89 | 1.00 ±0.00 / **0.95** ±0.00 | 303.02 ±0.01 / **294.49** ±0.20 |
| ResNet50 | **212.77** ±1.04 / 239.48 ±0.82 | 302.07 ±0.06 / **290.17** ±0.27 | 0.94 ±0.00 / **0.90** ±0.00 | **211.82** ±1.04 / 238.57 ±0.82 | 1.00 ±0.00 / **0.95** ±0.00 | 301.07 ±0.06 / **289.22** ±0.28 |
| ResNet101 | **212.55** ±0.76 / 239.27 ±0.57 | 301.96 ±0.07 / **289.44** ±0.32 | 0.94 ±0.00 / **0.90** ±0.00 | **211.59** ±0.76 / 238.36 ±0.57 | 1.00 ±0.00 / **0.95** ±0.00 | 300.96 ±0.07 / **288.49** ±0.32 |
| ResNet152 | **212.48** ±0.82 / 239.21 ±0.61 | 301.47 ±0.14 / **288.93** ±0.32 | 0.94 ±0.00 / **0.90** ±0.00 | **211.52** ±0.82 / 238.30 ±0.61 | 1.00 ±0.00 / **0.95** ±0.00 | 300.47 ±0.14 / **287.98** ±0.32 |
| ResNeXt101 | **209.14** ±0.53 / 236.20 ±0.46 | 301.40 ±0.04 / **288.46** ±0.36 | 0.94 ±0.00 / **0.90** ±0.00 | **208.18** ±0.53 / 235.29 ±0.46 | 1.00 ±0.00 / **0.95** ±0.00 | 300.40 ±0.04 / **287.51** ±0.36 |
| VGG16 | **210.64** ±1.29 / 237.56 ±1.04 | 303.26 ±0.15 / **293.20** ±0.59 | 0.93 ±0.00 / **0.88** ±0.00 | **209.70** ±1.29 / 236.65 ±1.04 | 1.00 ±0.00 / **0.95** ±0.00 | 302.26 ±0.15 / **292.25** ±0.59 |
| ShuffleNet | **216.70** ±0.82 / 243.02 ±0.65 | 304.02 ±0.05 / **295.74** ±0.28 | 0.94 ±0.00 / **0.90** ±0.00 | **215.74** ±0.82 / 242.11 ±0.65 | 1.00 ±0.00 / **0.95** ±0.00 | 303.02 ±0.05 / **294.79** ±0.28 |
| Inception | **217.32** ±0.85 / 243.58 ±0.73 | 304.09 ±0.03 / **297.24** ±0.76 | 0.95 ±0.00 / **0.90** ±0.00 | **216.36** ±0.85 / 242.67 ±0.73 | 1.00 ±0.00 / **0.95** ±0.00 | 303.09 ±0.03 / **296.29** ±0.76 |
| DenseNet161 | **212.15** ±0.94 / 238.91 ±0.75 | 302.02 ±0.02 / **289.26** ±0.53 | 0.94 ±0.00 / **0.90** ±0.00 | **211.19** ±0.94 / 238.00 ±0.75 | 1.00 ±0.00 / **0.95** ±0.00 | 301.02 ±0.02 / **288.31** ±0.53 |

*Table 10.* Observed p-value criteria across models. Hyperparameters are the same as Table 9. Each entry is reported as RAPS / PT-RAPS. Bold indicates the better, i.e., smaller, value between RAPS and PT-RAPS for each metric/model pair.

| Model | OU | OF | OM | OE |
|---|---|---|---|---|
| ResNet18 | 0.94 ±0.00 / **0.90** ±0.00 | **214.57** ±1.13 / 241.05 ±0.90 | 1.00 ±0.00 / **0.95** ±0.00 | 303.12 ±0.01 / **294.54** ±0.20 |
| ResNet50 | 0.95 ±0.00 / **0.90** ±0.00 | **212.37** ±1.03 / 239.06 ±0.81 | 1.00 ±0.00 / **0.95** ±0.00 | 301.17 ±0.06 / **289.27** ±0.27 |
| ResNet101 | 0.95 ±0.00 / **0.90** ±0.00 | **212.12** ±0.76 / 238.84 ±0.57 | 1.00 ±0.00 / **0.95** ±0.00 | 301.06 ±0.07 / **288.54** ±0.32 |
| ResNet152 | 0.95 ±0.00 / **0.90** ±0.00 | **212.06** ±0.82 / 238.79 ±0.61 | 1.00 ±0.00 / **0.95** ±0.00 | 300.58 ±0.13 / **288.03** ±0.31 |
| ResNeXt101 | 0.95 ±0.00 / **0.90** ±0.00 | **208.72** ±0.53 / 235.77 ±0.46 | 1.00 ±0.00 / **0.95** ±0.00 | 300.51 ±0.04 / **287.56** ±0.36 |
| VGG16 | 0.94 ±0.00 / **0.90** ±0.00 | **210.19** ±1.29 / 237.10 ±1.05 | 1.00 ±0.00 / **0.95** ±0.00 | 302.36 ±0.14 / **292.30** ±0.58 |
| ShuffleNet | 0.95 ±0.00 / **0.90** ±0.00 | **216.29** ±0.81 / 242.61 ±0.65 | 1.00 ±0.00 / **0.95** ±0.00 | 303.13 ±0.05 / **294.84** ±0.28 |
| Inception | 0.95 ±0.00 / **0.91** ±0.00 | **216.90** ±0.85 / 243.16 ±0.73 | 1.00 ±0.00 / **0.95** ±0.00 | 303.19 ±0.03 / **296.34** ±0.76 |
| DenseNet161 | 0.95 ±0.00 / **0.90** ±0.00 | **211.72** ±0.94 / 238.48 ±0.75 | 1.00 ±0.00 / **0.95** ±0.00 | 301.11 ±0.02 / **288.36** ±0.52 |

### D.1. Synthetic Datasets

This section presents experiment details about the motivating example (Section 3.2) in Appendix D.1.1 and failure case (Section 3.4) in Appendix D.1.2.

#### D.1.1. MOTIVATING EXAMPLE

In our motivating example, we consider a simple data-generating process where the true underlying model is linear with Gaussian mixture noise:

$$Y = \boldsymbol{X}^{\top}\boldsymbol{\beta} + \epsilon, \quad \boldsymbol{X} \sim \mathcal{N}(\mathbf{0}, I_2).$$

The noise term $\epsilon$ follows $\mathcal{N}(\mu, 1)$ with probability $0.5$ and $\mathcal{N}(-\mu, 1)$ with probability $0.5$. The training, calibration, and test folds are all generated from this distribution. To emulate model misspecification, we fit the training fold using a linear model with Gaussian noise. Throughout the experiments, we set $\mu = 20$, $\alpha \in \{0.1, 0.2\}$, and $p \in \{0.96, 0.98\}$. We further average results over 5 random seeds and report the corresponding standard errors. Both VCP (Algorithm 2) and PT-VCP are evaluated under this setting.

#### D.1.2. FAILURE CASE

In Figure 5, we present a failure case where VCP outperforms PT-VCP. Here, we modify the data-generating process to a linear model with Gaussian noise and fit it with the same linear model, so that no model misspecification arises in contrast to our motivating example. In this setting, the distribution of the score function fails to satisfy the sufficient condition in Theorem 3.10, which explains why VCP outperforms PT-VCP.

### D.2. Real World Datasets

In this section, we firstly introduce the model structure in our experiments in Appendix D.2.1. Then we present the experiment details, including experiments on marginal and group coverage (Appendix D.2.2, Appendix D.2.6), regression tasks (Appendix D.2.3), classification tasks (Appendix D.2.4), ablation studies (Appendix D.2.5) and interval stability (Appendix D.2.7).

*Table 11.* Comparison between CQR and PT-CQR regarding interval stability.

| Dataset | CQR | PT-CQR |
|---------|-----|--------|
| meps-19 | **0.00** ±0.000 | 0.13 ±0.005 |
| meps-20 | **0.00** ±0.000 | 0.13 ±0.006 |
| meps-21 | **0.00** ±0.000 | 0.14 ±0.003 |
| bike | **0.00** ±0.000 | 0.08 ±0.001 |
| blog-data | **0.00** ±0.000 | 0.11 ±0.003 |
| bio | **0.00** ±0.000 | 0.10 ±0.000 |
| facebook-1 | **0.00** ±0.000 | 0.10 ±0.001 |
| facebook-2 | **0.00** ±0.000 | 0.10 ±0.001 |
| concrete | **0.00** ±0.000 | 0.13 ±0.002 |
| star | **0.00** ±0.000 | 0.12 ±0.001 |

*Table 12.* Comparison between RAPS and PT-RAPS in classification tasks regarding interval stability, with $\alpha = 0.1, p = 0.95$ and index range chosen as 300.

| MODEL | BIAS | RAPS | PT-RAPS |
|-------|------|------|---------|
| RESNET18 | 40 | **0.02** ±0.004 | 12.08 ±0.060 |
| RESNET50 | 40 | **0.07** ±0.017 | 11.86 ±0.057 |
| RESNET101 | 40 | **0.01** ±0.004 | 11.93 ±0.089 |
| RESNET152 | 40 | **0.19** ±0.002 | 11.91 ±0.058 |
| RESNEXT101 | 40 | **0.20** ±0.000 | 11.89 ±0.083 |
| VGG16 | 40 | **0.11** ±0.030 | 12.00 ±0.060 |
| SHUFFLENET | 40 | **0.03** ±0.025 | 12.08 ±0.055 |
| INCEPTION | 40 | **0.07** ±0.010 | 12.19 ±0.080 |
| DENSENET161 | 40 | **0.03** ±0.006 | 11.90 ±0.057 |

### D.2.1. MODEL STRUCTURE

In this section, we present the details of the structure of our model on real world datasets. Specifically, our model shares the same structure as that of (Romano et al., 2019).

**Neural Net.** Our neural network design includes three fully connected layers, with ReLU activation functions applied between each layer. The initial layer accepts an input feature vector $X$ of n dimensions and produces 64 hidden units. The second layer mirrors this structure, generating another set of 64 hidden units. The final layer is a linear output layer that provides a pointwise prediction for the response variable $Y$. The network's parameters are optimized by minimizing a quadratic loss function. We used the Adam optimization algorithm with a constant learning rate of $5 \times 10^{-4}$, minibatch size of 64, and a weight decay coefficient of $10^{-6}$. In addition, regularization of dropout is implemented, with a retention probability of 0.1 for hidden units. To avoid overfitting, early stop is used and the number of training epochs is determined by cross-validation, with a maximum cap of 1000 epochs.

**CQR Neural Net.** We utilize neural networks to implement CQR for quantile regression. The network structure is consistent with the one described above, with the sole difference being that the output of the quantile regression network is a two-dimensional vector, which indicates the lower and upper conditional quantiles. Additionally, the training process remains the same, except that the pinball loss function is employed instead of the quadratic loss.

### D.2.2. MARGINAL AND GROUP COVERAGE

In Figure 1, we compare the coverage of VCP and PT-VCP on the BIKE dataset (Fanaee-T, 2013). Specifically, we evaluate several choices of $\alpha$ and $p$, and report both the marginal coverage and the group coverage (an empirical indicator of conditional coverage). The group coverage is obtained by partitioning the data according to Day, Month, and Year.

### D.2.3. REGRESSION TASKS

In ordinary regression tasks, we employ the neural network described in Section D.2.1 to fit several real-world datasets: MEPS19–21 (Cohen et al., 2009), BIKE (Fanaee-T, 2013), BLOG-DATA (Buza, 2014), BIO (Rana, 2013), FACEBOOK1–2 (Singh, 2015), CONCRETE (Yeh, 1998), and STAR (Achilles et al., 2008). To mimic model misspecification, we introduce a bias term that is directly added to the logits output by the neural network. The magnitude of this bias term, which varies

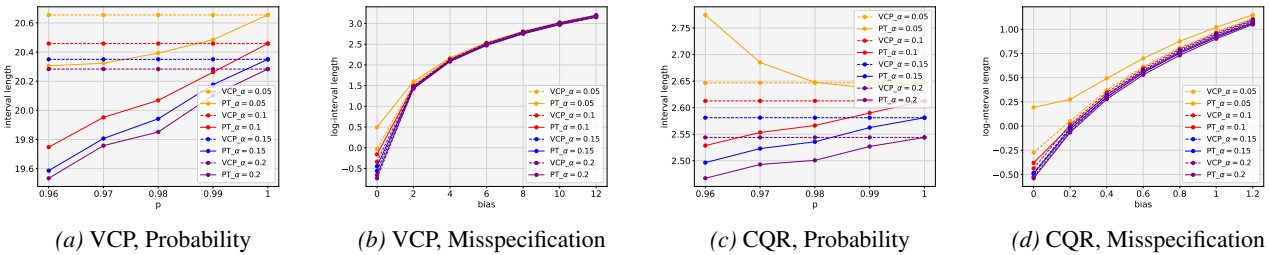

*(a)* VCP, Probability     *(b)* VCP, Misspecification     *(c)* CQR, Probability     *(d)* CQR, Misspecification

*Figure 6.* Ablation studies of dataset BIKE on different misspecification levels (b, d) and probability hyperparameters (a, c), including comparisons with VCP (a–b) and CQR (c–d).

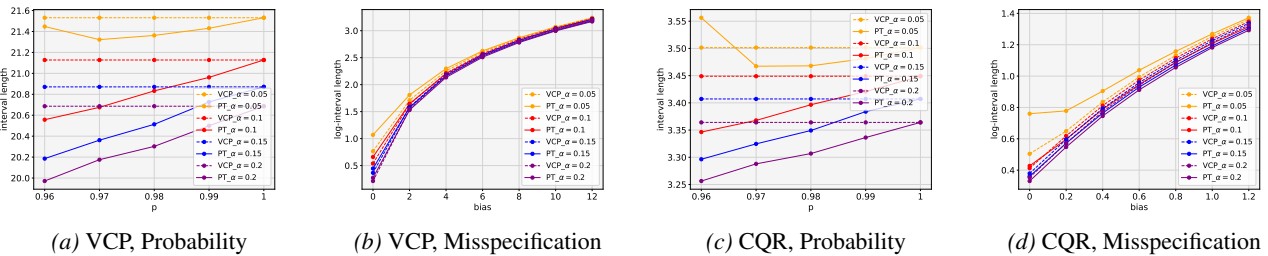

*(a)* VCP, Probability     *(b)* VCP, Misspecification     *(c)* CQR, Probability     *(d)* CQR, Misspecification

*Figure 7.* Ablation studies of dataset BIO on different misspecification level (a, c) and probability hyperparameter (b, d), including the comparison with VCP (a-b) and CQR (c-d).

across datasets, is reported in Table 2. Throughout these experiments, we set $\alpha = 0.1$ and $p = 0.95$, under which both VCP (Algorithm 2) and PT-VCP are evaluated. We further average results over 5 random seeds and report the corresponding standard errors.

In CQR tasks, we employ the CQR neural network described in Section D.2.1 to fit several real-world datasets: MEPS19–21 (Cohen et al., 2009), BIKE (Fanaee-T, 2013), BLOG-DATA (Buza, 2014), BIO (Rana, 2013), FACEBOOK1–2 (Singh, 2015), CONCRETE (Yeh, 1998), and STAR (Achilles et al., 2008). To mimic model misspecification, we introduce a bias term by directly adding it to both the lower and upper quantiles estimated by the quantile regression. The magnitude of this bias varies across datasets. Throughout the experiments, we set $\alpha = 0.1$ and $p = 0.95$, under which both CQR and PT-CQR are evaluated, as reported in Table 5. We further average results over 5 random seeds and report the corresponding standard errors.

### D.2.4. CLASSIFICATION TASK

In classification tasks, we apply PT to the real-world IMAGENET-VAL dataset (Deng et al., 2009) using several pre-trained models listed in Table 4, following a setting similar to Angelopoulos et al. (2020). To simulate model misspecification, we introduce a bias to the logits of several classes before the softmax operation. The magnitude of this bias is scaled according to the outputs, and the number of biased classes is specified by an index range in our experiments. For the classification task, we adopt RAPS (Angelopoulos et al., 2020) as the score function and set $\alpha = 0.1$ and $p = 0.95$. We further average results over 5 random seeds and report the corresponding standard errors.

### D.2.5. ABLATION STUDIES

The ablation studies mainly focus on regression tasks, including both ordinary regression and CQR. We conduct experiments on all datasets used in the regression setting. For these ablation studies, we set $\alpha \in \{0.05, 0.1, 0.15, 0.2\}$, $p \in \{0.96, 0.97, 0.98, 0.99, 1.00\}$, and apply dataset-specific bias magnitudes.

### D.2.6. GROUP COVERAGE

To demonstrate that PT-VCP does not degrade conditional coverage compared to VCP, we conduct experiments measuring group coverage on the MEPS19–21 (Cohen et al., 2009), BIKE (Fanaee-T, 2013), and STAR (Achilles et al., 2008) datasets.

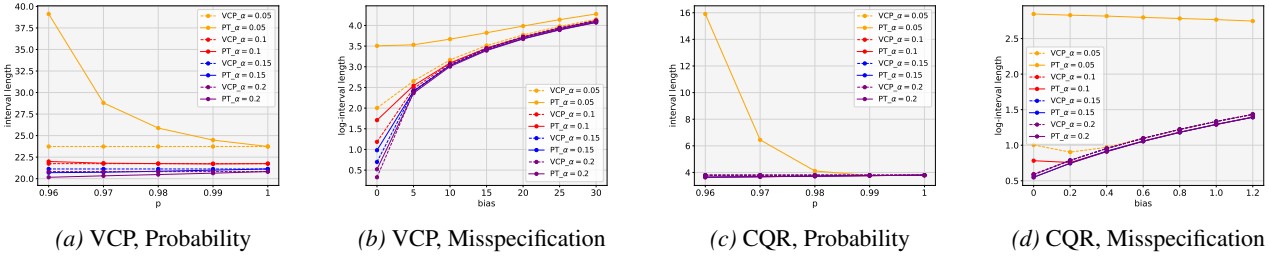

*(a)* VCP, Probability      *(b)* VCP, Misspecification      *(c)* CQR, Probability      *(d)* CQR, Misspecification

*Figure 8.* Ablation studies of dataset BLOGDATA on different misspecification level (a, c) and probability hyperparameter (b, d), including the comparison with VCP (a-b) and CQR (c-d).

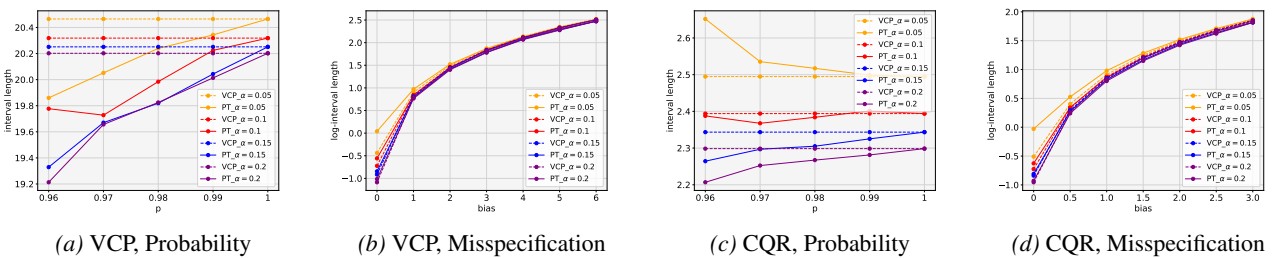

*(a)* VCP, Probability      *(b)* VCP, Misspecification      *(c)* CQR, Probability      *(d)* CQR, Misspecification

*Figure 9.* Ablation studies of dataset CONCRETE on different misspecification level (a, c) and probability hyperparameter (b, d), including the comparison with VCP (a-b) and CQR (c-d).

The grouping strategies are summarized in Table 8. In these experiments, we set $\alpha = 0.1$ and $p = 0.95$, and further average results over 5 random seeds, reporting the corresponding standard errors.

### D.2.7. INTERVAL STABILITY

To compare the newly proposed metric *interval stability* between VCP and PT-VCP, we conduct experiments on both regression and classification tasks, using different datasets and pre-trained models, respectively. The results are reported in Table 3, Table 11, and Table 12. We measure *interval stability* by computing the variance of the interval (or set) length (or size) for repeated predictions on the same input, and then averaging this variance over all inputs $X$. Results are further averaged over 5 random seeds, with the corresponding standard errors reported.

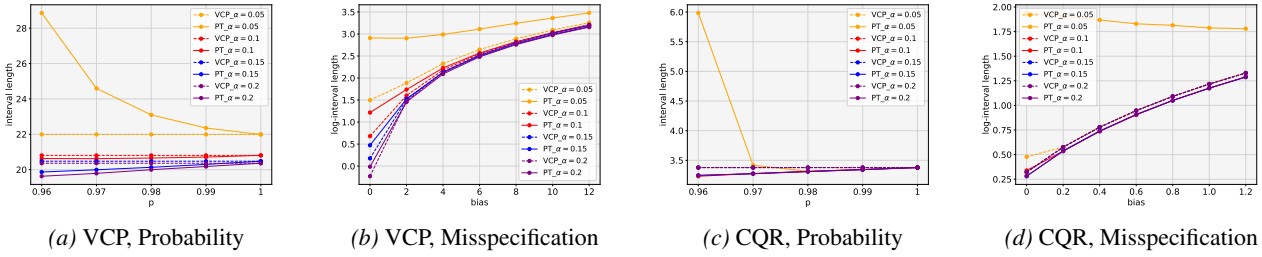

*(a)* VCP, Probability     *(b)* VCP, Misspecification     *(c)* CQR, Probability     *(d)* CQR, Misspecification

*Figure 10.* Ablation studies of dataset FACEBOOK1 on different misspecification level (a, c) and probability hyperparameter (b, d), including the comparison with VCP (a-b) and CQR (c-d).

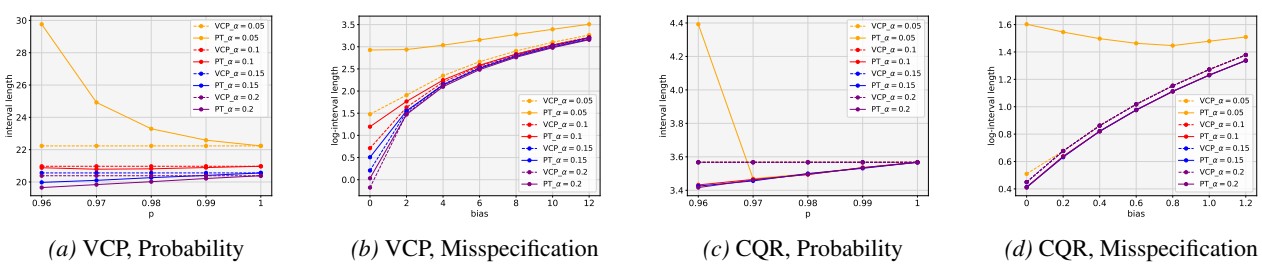

*(a)* VCP, Probability     *(b)* VCP, Misspecification     *(c)* CQR, Probability     *(d)* CQR, Misspecification

*Figure 11.* Ablation studies of dataset FACEBOOK2 on different misspecification level (a, c) and probability hyperparameter (b, d), including the comparison with VCP (a-b) and CQR (c-d).

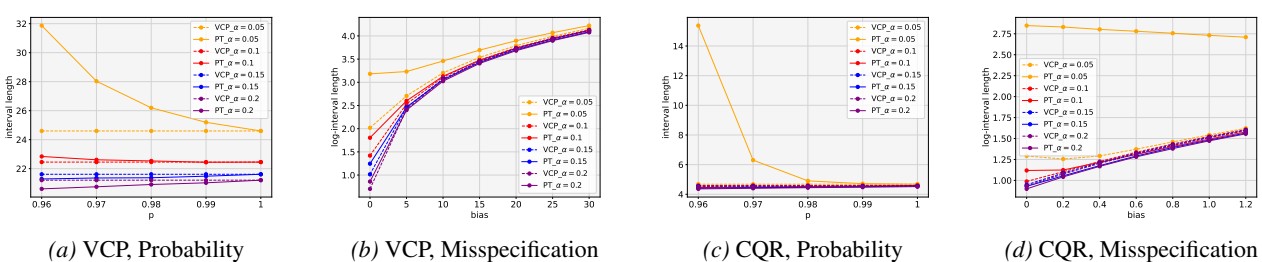

*(a)* VCP, Probability     *(b)* VCP, Misspecification     *(c)* CQR, Probability     *(d)* CQR, Misspecification

*Figure 12.* Ablation studies of dataset MEPS19 on different misspecification level (a, c) and probability hyperparameter (b, d), including the comparison with VCP (a-b) and CQR (c-d).

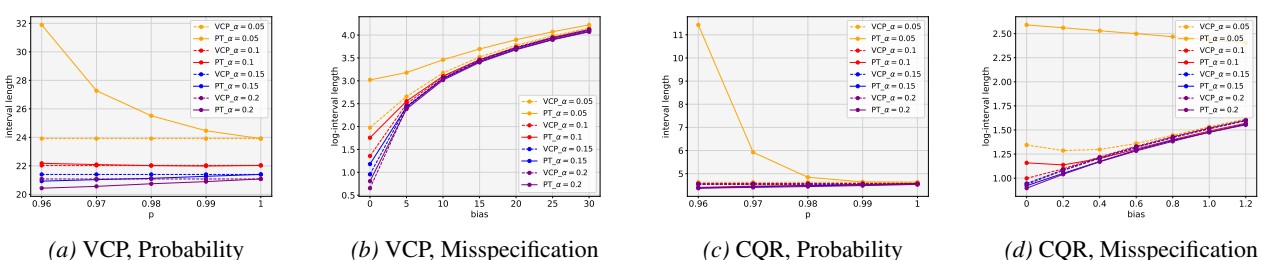

*(a)* VCP, Probability     *(b)* VCP, Misspecification     *(c)* CQR, Probability     *(d)* CQR, Misspecification

*Figure 13.* Ablation studies of dataset MEPS20 on different misspecification level (a, c) and probability hyperparameter (b, d), including the comparison with VCP (a-b) and CQR (c-d).

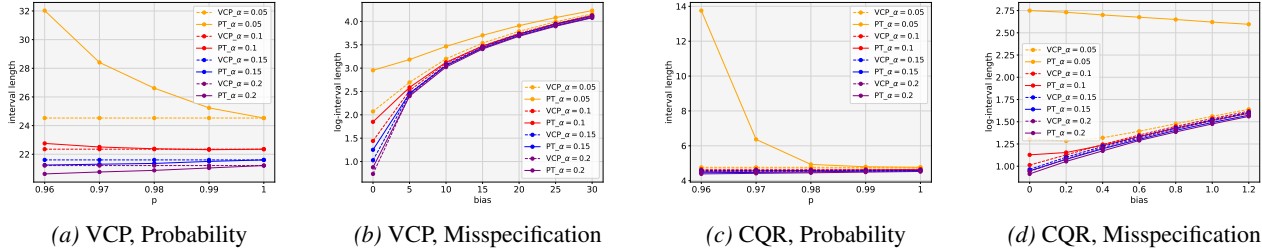

*(a)* VCP, Probability      *(b)* VCP, Misspecification      *(c)* CQR, Probability      *(d)* CQR, Misspecification

*Figure 14.* Ablation studies of dataset MEPS21 on different misspecification level (a, c) and probability hyperparameter (b, d), including the comparison with VCP (a-b) and CQR (c-d).

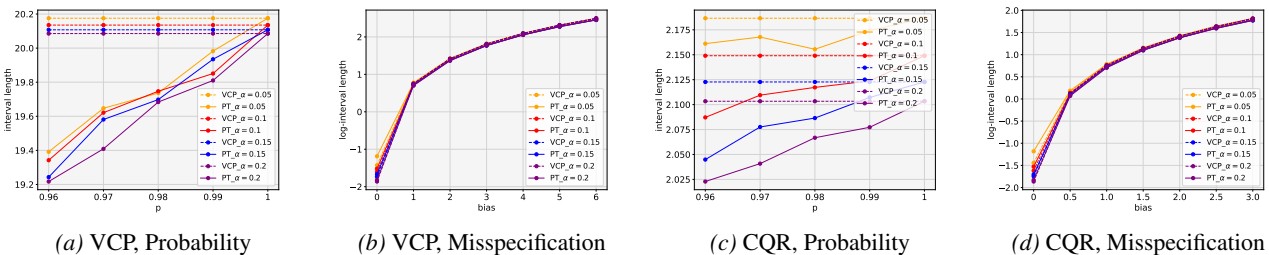

*(a)* VCP, Probability      *(b)* VCP, Misspecification      *(c)* CQR, Probability      *(d)* CQR, Misspecification

*Figure 15.* Ablation studies of dataset STAR on different misspecification level (a, c) and probability hyperparameter (b, d), including the comparison with VCP (a-b) and CQR (c-d).

