# OpenReview forum: "Questioning the Coverage-Length Metric in Conformal Prediction: When Shorter Intervals Are Not Better"
_ICML.cc/2026/Conference — ICML 2026 regular_

### Official Review · Reviewer_mp8c · 2026-03-02

**Soundness:** 3
**Presentation:** 2
**Significance:** 4
**Originality:** 4
**Overall Recommendation:** 5
**Confidence:** 5

**Summary:**

The paper introduces the notion of prejudicial trick. This is when we randomize between reporting a bigger interval length and a set of length 0. This can some time reduce the expected interval length while keeping the coverage intact.
This raises a concern that modern conformal prediction research may contain some component of this trick to beat the benchmark on coverage and length.

**Compliance With Llm Reviewing Policy:**

Affirmed.

**Final Justification:**

The author addresses my concern.

**Key Questions For Authors:**

1. Why would you have theorem 3.6? What is the purpose of it?
2. Where do you use IS? Is it measured in any experiments?
3. I think explaining the weakness can also be a question itself.

**Limitations:**

yes

**Strengths And Weaknesses:**

Strengths:
1. PT is a very nice and simple algorithm. We don't need to know anything about the structure of the problem, and only require mixture of coverages, so that its expected value is the same as the target. [at least as long as we don't care whether the length will be reduced in expectation or not].
2. It's good that the paper has a small example in the beginning. However, after reading through the paper (sufficiency conditions), this also makes me doubt whether this work because of a very specific structure of having discrete numbers responsible for this and whether it is really hard to get in a continuous setting.
3. The proposed definition of interval stability is nice. The paper has emphasized before that the unfairness is not about the one across the inept space, but just the one across different realization of the algorithm. This is nicely captured in the definition.

Weaknesses:
1. The theorem organization is bad.
1.1 The theorem 3.3 and 3.4 can be merged into a single theorem. Having them separately does not give any more insights and it is also quite obvious for both theorem.
1.2 The theorem 3.6 is just a fluff. There is no material in that.
1.2.1 The theorem just introduce a confusing function called $\mathcal{F}(p)$. This function only serves in the theorem; there is no discussion on what the interpretation of what this function means or what can be a sufficient condition for the so called sufficient condition of $\mathcal{F}(1)-\mathcal{F}(p) \ge 1-p$ to hold.
1.2.2 If one decodes this theorem carefully (or look at its proof in the appendix), the theorem literally says that
``if the PT coverage is weakly better than CP coverage, then the PT coverage is weakly better than CP coverage". The entire $\mathcal{F}(1)-\mathcal{F}(p) \ge 1-p$ just means that and nothing else. One could even call it a necessary condition.
1.2.3 I personally think that what I list as 1.2.1 is just a minor error but can make the reader confusing, but the issue in 1.2.2 is the worst. The theorem is correct, but it does not give any more insights apart from being a waste of time and space. It is just a play with word and mathematical notation.
1.3 Lemma 3.7 is also the similar. It just says that if length of PT is lower than length of CP, then length of PT is lower than length of CP. This is more like a recharacterization (unless we assume that the null set can have non-zero probability, which can happen when there is atomic distribution in regression task.)
1.4 What is corollary 3.11 a corollary of? It cannot be a corollary of theorem 3.8 since theorem 3.8 is less narrow. If it is a corollary from the proof of theorem 3.8, it should be stated. If it is a corollary of lemma 3.7, then theorem 3.8 will also be a corollary.
2. I really like the main idea of this paper, but it could have been delivered in a much more concise manner. The idea does not have to be lengthy to be good and this paper is like that. The mathematics underlying this phenomenon is simple, short, and nice. However, the author seems to forcefully add these meaningless theory components, especially on sufficiency when nothing can be translated into a practical test (apart from corollary 3.11). The paper has not even tried to show some example when this problems can exist, and only show that it does not exist in gaussian case. The paper will benefit if these spaces are dedicated to experiments.

Suggestion:
1. The paper really should emphasize that this phenomenon can easily happen when we have classification problem. Whenever the smallest coverage given by vanilla conformal prediction exceed the target of $1-\alpha$, we can simply do the randomization for it to be null with some probability so that the expected coverage falls to the target level of $1-\alpha$
2. Theorem 3.8 is indeed string since it requires differentiability of L in a pathwise manner. However, in most cases, if we move the expectation to be inside, so we take derivative of expectation instead of expectation of derivative, it is more likely that this can be well-defined, given that the steps will smoothed out from the expectation over x [See https://arxiv.org/abs/2602.05119 (Sornwanee 2026)] By doing so, you can make a toy example of simple model, like linear regression and show the condition when the result holds. In fact, corollary 3.9 is a sufficient condition even when the condition of theorem 3.8 does not hold.
3. Example 3 needs more description. I think it probably just a result that the pdf is decreasing from the center. Therefore, a similar result will also be seen in any symmetric unimodal distribution.

Small suggestion:
1. in algorithm 1, I think it will be easier say (in line 4) that $C_{1-\alpha}(x') = \{\hat{\mu}(x')\}$ to emphasize that it is an interval (but of 0 length).

---

> ### Author Rebuttal · Authors · 2026-03-30
>
> We thank the reviewer for the careful reading and for recognizing the simplicity of PT and the usefulness of Interval Stability. We also thank the reviewers for their suggestions on the expression and content organization of the paper. We address these concerns below.
>
> **1. On theorem organization** (Weakness 1 & Q1).
>
> We agree that Section 3 should be streamlined. Theorems 3.3 and 3.4 can be presented more compactly, since their main role is to show that PT does not improve length by merely sacrificing coverage. We also agree that Lemma 3.7 is closer to a recharacterization of the master inequality and is best viewed as a bridge criterion rather than a main conceptual result.
>
> Regarding Theorem 3.6, its role is only to show that PT may also appear favorable under subset/group conditional-coverage comparisons even when the base method does not satisfy exact conditional coverage. The reviewer is also right that the current function $\mathcal{F}(p)$ is confusing; in the current draft we do not provide an interpretable sufficient condition for $\mathcal{F}(1)-\mathcal{F}(p)\ge 1-p$. In this sense, the result is closer to a characterization than to an insightful standalone theorem, and we will de-emphasize it.
>
> The reviewer is right that the dependency of Corollary 3.11 is currently unclear. Corollary 3.11 is better viewed as an alternative sufficient condition derived from the same general criterion underlying Lemma 3.7, rather than as a direct corollary of Theorem 3.8. Theorem 3.8 gives the differentiable first-order version, while Corollary 3.11 gives the non-differentiable secant version. We will revise this.
>
> **2. On positive examples in continuous settings** (Weakness 2).
>
> We agree that the current draft makes the positive continuous case less transparent than the Gaussian failure case. Our intent, however, was not to suggest that PT only succeeds in discrete settings. Empirically, the paper already contains successful regression cases (Example 2 and Table 2), and  Theorem 3.8, Corollary 3.9, and Corollary 3.11 provide sufficient conditions for continuous settings.
>
> We also agree that this part would benefit from a simple toy illustration. In revision, we will restate the first-order sufficient condition directly in terms of the averaged length function $G(u)=\mathbb{E}[L(x,u;s)]$, and add a simple continuous toy regression example with U-shaped noise, where $G(u)=2\sqrt{u}$, so $G$ is strictly concave and PT yields $pG(u/p)=2\sqrt{up}<2\sqrt{u}=G(u)$ for any $p\in(u,1)$. This gives a simple continuous setting in which PT strictly reduces expected length.
>
> **3. On classification task** (Suggestion 1).
>
> We agree that the classification case makes the phenomenon especially transparent. Whenever the base conformal predictor is conservative and attains coverage above the target, one can randomize between the original prediction set and the empty set to match the target coverage in expectation while strictly reducing the average set size. We will make this intuition more explicit in the revision.
>
> **4. On Theorem 3.8** (Suggestion 2).
>
> We found the reviewer’s suggestion on Theorem 3.8 very helpful. We agree that the current pathwise differentiability assumption is stronger than necessary as stated, and we will reformulate the first-order sufficient condition directly in terms of $G(u)=\mathbb{E}[L(x,u;s)]$.
>
> **5. On Example 3** (Suggestion 3).
>
> We also agree that Example 3 deserves more explanation. Its purpose is to provide a clean failure case showing that PT is not universally beneficial, rather than to suggest that failure is specific to the Gaussian case. We agree that the proof likely reflects a broader geometric phenomenon and may extend to broader symmetric unimodal score distributions.
>
> **6. On Interval Stability and experiments** (Q2).
>
> Interval Stability is evaluated in Table 3 in the main paper, and additionally in Table 7 (for CQR) and Table 8 (for classification). We agree that these results should be made much more visible in the revision.
>
> **7. On the notations** (Small Suggestion).
>
> We also thank the reviewer for the notation suggestion in Algorithm 1, and we agree that writing $C_{1-\alpha}(x')=\{\hat{\mu}(x')\}$​ in regression makes it clearer that the output is still a prediction set (a singleton set of zero length).
>
> Overall, we agree that the paper should present its message more directly and concisely. In revision, we will streamline Section 3 by merging or de-emphasizing mainly structural results (including Theorems 3.3–3.4, Lemma 3.7, and Theorem 3.6), clarify the status of Corollary 3.11, add discussion about classification case in Section 3.4, restate the first-order condition in terms of the averaged length function in Theorem 3.8, add a simple continuous toy success case in Section 3.4, expand the discussion of Example 3, and highlight the existing Interval Stability experiments more clearly in the main text in Section 4.

---

> > ### Author Rebuttal · Reviewer_mp8c · 2026-03-31
> >
> > Thank you.
> >
> > Overall, I think that the paper message is nice and clear, and I will be willing to increase the score as long as the author can address the problem of theoretical part, which is, at the moment, not incorrect but mostly useless.

---

> > > ### Author Response · Authors · 2026-04-03
> > >
> > > Thank you again for the positive update and for the very careful feedback. We are glad that our rebuttal clarified the main points. We will revise the paper accordingly. In the new version, the theory part will be made substantially leaner and more transparent: Theorems 3.3 and 3.4 will be merged, the current Lemma 3.7 will be turned into a short reduction step in Section 3.4 based on $G(u)=E[L(x,u;s)]$, and Theorem 3.6 will be heavily de-emphasized. We will also clarify the status of Corollary 3.11, restate the first-order sufficient condition directly in terms of $G(u)$, add a simple continuous toy success case, and make the classification case and existing Interval Stability experiments more visible in the main text. Overall, the final paper will keep the same message, but present it in a much more concise and direct way.

---

### Official Review · Reviewer_xjbL · 2026-03-12

**Soundness:** 4
**Presentation:** 4
**Significance:** 4
**Originality:** 4
**Overall Recommendation:** 5
**Confidence:** 4

**Summary:**

Conformal prediction methods are usually assessed through the coverage–length tradeoff: for a fixed target coverage level, methods that produce smaller prediction sets are generally considered better. The paper introduces the Prejudicial Trick (PT), a counterintuitive wrapper around any conformal prediction algorithm, and shows that it can (1) retain valid marginal coverage, (2) retain conditional coverage whenever the base method has it, (3) under certain conditions, reduce average prediction set size, and (4) empirically achieve smaller average set sizes on real-world datasets.

**Compliance With Llm Reviewing Policy:**

Affirmed.

**Final Justification:**

The authors answered my questions clearly and satisfactorily, and I remain positive on the paper's soundness, clarity, and practical relevance.

**Key Questions For Authors:**

Interval Stability seems well suited to detect PT-like randomness. Do the authors have any intuition for whether, in the conditional coverage setting, the coverage–length metric could still be misleading even when Interval Stability is zero, for example under deterministic constructions?

**Limitations:**

yes

**Strengths And Weaknesses:**

### Strengths
* The paper offers a novel perspective and a valuable insight for the conformal prediction community, with meaningful practical implications. In particular, it makes a convincing case that conformal prediction methods involving randomness in the construction of prediction sets should also be evaluated using the proposed Interval Stability metric, to guard against unintentional coverage-length hacking.
* The theoretical results appear sound and thorough.
* The paper is very well written, well structured, and easy to follow; the examples are clear and intuitive.
* The empirical results on real-world benchmarks strengthen the paper by showing that the issue is not merely a pathological theoretical possibility.

### Weaknesses
* The paper does not identify a concrete previously published conformal prediction method whose reported coverage–length performance is empirically shown to rely on a PT-like effect. Nevertheless, the paper remains important, since the theoretical possibility alone is already meaningful, and similar effects could plausibly arise unintentionally in existing conformal prediction methods.

---

> ### Author Rebuttal · Authors · 2026-03-30
>
> We thank the reviewer for the positive assessment of our paper and for recognizing its technical soundness, practical relevance, and theoretical support. We are especially encouraged that the reviewer agrees with the main message of the paper: the standard coverage-length evaluation can be misleading in the presence of randomness, and Interval Stability provides a useful additional diagnostic against such effects. We also appreciate the reviewer’s question about the zero-stability regime, since it helps sharpen the scope of our contribution.
>
> **1. On the conceptual contribution of PT** (Weakness 1).
>
> We thank the reviewer for recognition of the importance of our work. We agree that the current paper does not identify a previously published conformal prediction method whose empirical gains can be definitively traced to a PT-like effect. We view this as a limitation of the present empirical scope, but not of the conceptual contribution. Our goal in this work is to present a minimal constructive counterexample showing that the standard coverage-length metric can be manipulated even without using task-specific information. The main contribution is therefore to establish that this loophole is real, that it can arise from randomized prediction-set construction, and that it should be guarded against when evaluating conformal methods.
>
> **2. On the coverage-length metric under IS=0 regimes** (Q1).
>
> We thank the reviewer for this thoughtful question. Our intuition is as follows. If Interval Stability is zero, then the specific PT-like mechanism studied in our paper is absent at the level of interval size. Specifically, $IS = 0$ implies that, conditional on a fixed $(x, D_{\mathrm{cal}})$, the returned interval size does not vary across runs. This does not necessarily mean that the whole method is fully deterministic; rather, it means that any remaining randomness is size-preserving and therefore cannot create the same apparent gain in average length through run-to-run randomization as PT. At the same time, if one evaluates only coverage and average length, then misleading behavior may still arise under deterministic constructions when conditional reliability is poor. However, once conditional coverage is also taken into account, our current paper does not claim a further problem of the same kind. In this sense, our point is complementary to conditional coverage that PT shows that even when conditional coverage is preserved, randomness in prediction-set construction can still make the standard coverage-length evaluation look deceptively favorable. This is precisely why Interval Stability is useful as an additional diagnostic.
>
> We will revise the paper to make this boundary explicit. In particular, we will clarify that Interval Stability is designed to diagnose randomness-induced instability; that zero Interval Stability rules out the specific PT-like effect studied in this work in Section 4. We thank the reviewer again for the highly positive evaluation and for raising this important point, which will help us sharpen the presentation of the paper.

---

> > ### Author Rebuttal · Reviewer_xjbL · 2026-04-02
> >
> > The authors answered my questions clearly and satisfactorily, and I remain positive on the paper's soundness, clarity, and practical relevance.

---

> > > ### Author Response · Authors · 2026-04-03
> > >
> > > We thank the reviewer for the positive assessment and for recognizing that our rebuttal have addressed the concerns. We are glad that the clarification and additional discussion resolved the issues raised earlier. The corresponding changes have been incorporated into the revised manuscript. We appreciate the reviewer’s time and thoughtful feedback.

---

### Official Review · Reviewer_nx5b · 2026-03-13

**Soundness:** 4
**Presentation:** 3
**Significance:** 3
**Originality:** 3
**Overall Recommendation:** 5
**Confidence:** 3

**Summary:**

The paper exposes an interesting flaw in the expected coverage-length metric that is used to evaluate conformal prediction intervals. The authors introduce the "Prejudicial Trick" (PT) which is an input-independent randomized procedure that assigns a null interval ($\emptyset$) with probability $1-p$ and a tighter CP interval (with miscoverage $\alpha' = 1 - \frac{1-\alpha}{p}$) with probability $p$. By construction, PT satisfies marginal and conditional validity, but under common conditions frequently induced by model misspecification, it strictly decreases the expected average interval length. To detect such deceptive randomized length reductions, the authors propose a new metric, "Interval Stability," defined as the conditional variance of the interval measure.

**Compliance With Llm Reviewing Policy:**

Affirmed.

**Final Justification:**

The authors successfully address all my concerns.

**Key Questions For Authors:**

- How does the Interval Stability metric distinguish between the deceptive randomness of PT and the benign randomness of modern complex base models (e.g., Monte Carlo dropout or deep ensembles)?
- How does the Interval Stability metric mathematically decouple intrinsic data variance (e.g., in highly non-stationary distributions or under severe covariate shift) from the extrinsic algorithmic randomness it is explicitly designed to penalize?
- Your analysis is tied to a global p. What happens when p is feature-dependent? Could there be a smarter adversary that changes p with the features X such that even the Interval Stability metric is fooled?

I am very willing to reconsider my score if these points are addressed.

**Limitations:**

No. The authors must revise the Impact Statement to explicitly discuss fairness and risks of deploying PT in practice. While presented as an adversarial auditing tool, the paper exposes a mechanism that could severely harm algorithmic fairness if replicated in high-stakes decision-making domains.

**Strengths And Weaknesses:**

- The theoretical claims are mathematically rigorous and well-supported. The empirical validation across diverse datasets and base algorithms also corroborates the theory. A minor weakness is the lack of theoretical analysis of PT under non-exchangeable conditions (e.g., covariate shift), though the i.i.d. derivations remain fully sound.

- The paper is clearly written. Example 1 helps build intuition for the adversarial mechanism. I'd suggest exploring the connection between PT and Localized CP (Proposition 3.2) more in the main text.

- Proving that the naive length metric can be manipulated implicitly or explicitly is impactful to revert to more robust metrics to evaluate UQ methods. The authors also propose an immediate fix. However, it is unclear how the "fix" discriminates between the intrinsic randomness in data/models vs adversarial stochasticity.

- While the impossibility of exact conditional coverage already known (e.g., Barber et al., 2021), weaponizing input-independent randomness to strictly reduce the marginal interval length is a novel adversarial approach. The connection to model misspecification via localized concavity is also creative and impactful.

---

> ### Author Rebuttal · Authors · 2026-03-30
>
> We thank the reviewer for the careful reading and the positive assessment of the paper’s soundness, clarity, and originality. We are encouraged that the reviewer views the core technical contribution as solid, and we appreciate the questions raised here, which mainly concern the scope and interpretation of Interval Stability, as well as the role of feature-dependent adversaries and fairness considerations. In response, we aim to clarify these boundaries more explicitly and to make the corresponding revisions in the manuscript more concrete.
>
> **1. On distinguishing randomness** (Q1).
>
> We thank the reviewer for this thoughtful question. We would like to clarify that Interval Stability is not intended to distinguish whether a source of randomness is “benign” or “malicious”. Our point is that any algorithmic randomness that can contribute to an apparent reduction in average length should be made visible when evaluating a method. Accordingly, if methods such as MC dropout or deep ensembles produce different prediction-set sizes across repeated runs for the same input and calibration set, Interval Stability will reflect that variability. The role of the metric is not to judge the origin of the randomness, but to reveal when reported efficiency gains may depend on run-to-run randomness.
>
> **2. On how intrinsic data variance affects interval stability** (Q2).
>
> Our Interval Stability is defined by first taking the variance over algorithmic randomness conditional on a fixed test point and calibration set, and then taking the expectation over the data distribution. As a result, variability arising purely from heterogeneity across inputs does not itself contribute to Interval Stability. In particular, intrinsic differences across regions of the feature space, such as heteroscedasticity or non-stationarity, may cause a method to return wider intervals in more difficult regions, but they are not counted as instability unless there is also run-to-run variability for the same fixed $(x, D_{cal})$. Therefore, a method may exhibit substantial variation in interval width across inputs while still having zero Interval Stability.
>
> **3. On the feature-dependent $p$** (Q3).
>
> Our current analysis focuses on a global $p$ because it gives the simplest constructive counterexample. If one allows a feature-dependent $p(x)$, the current conditional-coverage proof does not directly apply, and we will make this scope explicit. However, feature dependence does not by itself fool Interval Stability: if the adversary remains genuinely randomized at each fixed $x$, then the resulting same-input run-to-run variability is still detected by Interval Stability. The metric can only be fooled if the manipulation becomes deterministic at each fixed input, in which case the issue is no longer the randomness-induced problem studied in this paper, but rather a different subgroup/feature-allocation problem.
>
> **4. Revise plan**
>
> We also agree that these issues should be addressed concretely in the revised manuscript. Specifically, in Section 4 we will revise the presentation around Definition 4.1 to emphasize directly from the definition that Interval Stability first takes variance over algorithmic randomness conditional on a fixed $(x, D_{\mathrm{cal}})$, and only then averages over the data distribution; this will make clear that input heterogeneity itself is not counted as instability. In Section 3.2, we will add a short discussion to clarify that a feature-dependent $p(x)$ is a natural extension of PT but is outside the scope of the present manuscript. We will add a discussion in Section 4, after Proposition 4.2, to clarify that feature dependence by itself does not evade Interval Stability if the construction remains randomized for a fixed $(x,D_{cal})$. We will also expand the discussion of Proposition 3.2 and Remark 1.3 so that the connection between PT and localized CP is explained more transparently, rather than appearing only as a brief side remark. Finally, we will revise Impact Statement, together with Section 3.5 to spell out the fairness and deployment risks more explicitly and any constructor similar to PT should be avoided.

---

> > ### Author Rebuttal · Reviewer_nx5b · 2026-04-04
> >
> > I thank the authors for their detailed response. This addresses all my concerns and I am happy to raise my score. All the best :)

---

> > > ### Author Response · Authors · 2026-04-04
> > >
> > > We are sincerely grateful to the reviewer for the positive feedback and for raising the score. We are pleased to hear that our detailed response addressed the concerns effectively. We also appreciate the reviewer's kind wishes and thoughtful evaluation throughout the process. All the feedback has been carefully integrated into the revised manuscript to further improve its quality.

---

### Official Review · Reviewer_jobk · 2026-03-22

**Soundness:** 3
**Presentation:** 3
**Significance:** 2
**Originality:** 3
**Overall Recommendation:** 3
**Confidence:** 3

**Summary:**

This paper examines the standard evaluation of conformal prediction methods based on coverage and average interval length. It introduces a simple construction, called the Prejudicial Trick (PT), which preserves marginal coverage while reducing average interval length by randomly returning null predictions for a subset of test points and compensating with more conservative intervals elsewhere. The authors provide theoretical conditions under which PT achieves these improvements and validate the effect empirically across several datasets. Despite appearing favorable under the coverage–length metric, PT produces unstable and uneven predictions across samples. To address this issue, the paper proposes interval stability as an additional diagnostic metric to detect such behavior. Overall, the work highlights a limitation of relying solely on coverage and average length for evaluating CP methods.

**Compliance With Llm Reviewing Policy:**

Affirmed.

**Key Questions For Authors:**

1. The analysis focuses on average set size. Does the PT construction still appear advantageous under other efficiency criteria (e.g., p-value–based or observed metrics in [1])? If not, to what extent is the issue specific to this particular metric?

2. The conditional coverage improvement is defined by averaging over algorithmic randomness. How does PT behave at the level of individual predictions (i.e., for a fixed random draw)? Can PT lead to systematic failures on a subset of inputs despite satisfying conditional coverage in expectation?

3. The paper suggests that real CP methods may implicitly exhibit PT-like behavior. Can the authors provide concrete examples or formal conditions under which commonly used methods (e.g., CQR or score-based methods like APS/RAPS) exhibit similar effects?

4. Is interval stability intended as a general evaluation criterion or only as a diagnostic for PT-like constructions? How informative is this metric for standard CP methods that are largely deterministic or have limited randomness, such as CQR or score-based methods like APS/RAPS?

5. The length improvement relies on specific conditions (e.g., misspecification or concavity assumptions). How robust are the conclusions when these conditions do not hold? Can the authors characterize when PT fails to improve efficiency?

**Limitations:**

Yes

**Strengths And Weaknesses:**

**Strengths**

The paper presents a constructive counterexample (PT) showing that the standard coverage–length evaluation in CP can be manipulated. The construction is simple and supported by both theoretical analysis and empirical results. The analysis of conditional coverage highlights a subtle gap between marginal guarantees and practical behavior. The introduction of interval stability provides a useful diagnostic perspective for detecting instability induced by randomized constructions.

**Weaknesses**

The scope of the conclusions is limited to the coverage–length metric, and the paper does not examine whether similar issues arise under alternative efficiency criteria studied in prior work [1]. The claimed improvement in conditional coverage is based on averaging over algorithmic randomness and does not reflect per-instance reliability, which may lead to a misleading interpretation of practical performance. The connection between the PT construction and real CP methods is suggestive but not rigorously established. The empirical evaluation focuses mainly on coverage and length, without analyzing per-instance behavior or other evaluation metrics. Finally, the proposed interval stability is presented as a diagnostic tool but is not studied as a general evaluation criterion, and its applicability beyond the PT setting remains unclear. More broadly, it is unclear whether this issue reflects a fundamental limitation or a metric-specific artifact.

[1] Vovk, Vladimir, et al. "Criteria of efficiency for conformal prediction." Symposium on conformal and probabilistic prediction with applications. Cham: Springer International Publishing, 2016.

---

> ### Author Rebuttal · Authors · 2026-03-29
>
> We thank the reviewer for the thoughtful feedback and we are glad that the reviewer recognizes the value of our work as a constructive counterexample and the usefulness of interval stability. We understand the reviewer’s main concern is about whether our findings reflect a fundamental limitation or are specific to the coverage–length metric. We would like to clarify that our primary goal is to critically examine the coverage–length evaluation paradigm, which is the most widely used standard in conformal prediction. Our results show that even this widely adopted combination can be systematically manipulated. Therefore, our contribution is not to claim a universal limitation of all evaluation criteria, but to demonstrate that the current reliance on coverage and average interval length can be insufficient for assessing practical performance. Given how central this evaluation protocol is in the CP literature, we believe this limitation is important in its own right. We provide additional experiments at: https://anonymous.4open.science/r/icml2026-rebuttal-8440-AC7E/Reviewer_jobk.md and address the concerns below.
>
> **1. On how PT behaves on other criteria** (Q1).
>
> We thank the reviewer for this important question. To assess whether the issue is metric-specific, we conducted additional experiments using the efficiency criteria in Criteria of Efficiency for Conformal Prediction. Since these are defined for classification, we evaluated RAPS and PT-RAPS across all ten criteria.
>
> We find that PT outperforms the baseline on 7 out of 10 metrics (Table A and Table B in the link), suggesting that the phenomenon is not limited to average length. Instead, PT can appear favorable under multiple commonly used criteria, despite exhibiting instability and uneven per-instance reliability.
>
> **2. On per-instance behavior** (Q2).
>
> We agree that per-instance behavior is crucial. For a fixed realization of randomness, PT can indeed fail systematically on a subset of inputs (e.g., returning empty sets), despite satisfying conditional coverage in expectation. This is precisely the issue we highlight which is that **averaging over algorithmic randomness can mask poor per-instance behavior**. Thus, PT may appear valid under expectation-based metrics while being unreliable for individual instances, motivating metrics such as interval stability.
>
> **3. On the connection to existing CP methods** (Q3).
>
> We thank the reviewer for this important question. We clarify that our claim is not that existing methods explicitly implement PT, but that randomness in practical CP pipelines can implicitly induce PT-like effects. In Proposition 3.2, we show that PT can be viewed as a special case of localized CP under a degenerate distribution of the scale estimator $\hat{\sigma}(x)$. Localized CP uses the normalized score $\hat{S}^{\text{norm}}(x,y) = \hat{S}(x,y)/\hat{\sigma}(x)$, where $\hat{\sigma}(x)$ is learned via stochastic procedures (e.g., neural networks or random forests), and is therefore random across runs. When $\hat{\sigma}(x)$ follows a two-point distribution, localized CP reduces exactly to PT.
>
> To further bridge theory and practice, we also consider a relaxed variant where $\hat{\sigma}(x)=0$ is replaced by a small constant $\epsilon$, which avoids null sets while exhibiting similar behavior (Table C in the link). This makes the connection closer to practical localized CP settings.
>
> **4. On interval stability as an evaluation metric** (Q4).
>
> We view interval stability as a general complementary evaluation criterion, not specific to PT. While PT illustrates an extreme case, the metric applies to any method with stochastic components. Even for largely deterministic methods (e.g., CQR, RAPS), variability can arise from training randomness. Our appendix results (Tables 3, 7, 8) show that interval stability provides additional insights beyond coverage and length. We also add experiment of the interval stability of localized CP where $\hat{\sigma}(x)$ is trained several times and the nerual net we use to train $\hat{\sigma}(x)$ is the same as [1], please see results at Table D in the link.
>
> [1] Seedat N, Jeffares A, Imrie F, et al. Improving adaptive conformal prediction using self-supervised learning[C]//International Conference on Artificial Intelligence and Statistics. PMLR, 2023: 10160-10177.
>
> **5. On robustness and limitations of PT** (Q5).
>
> We emphasize that our conditions are sufficient but not necessary. PT can still possibly yield improvements beyond these settings, but may fail when the model is well-specified or under minimal covariate shift. Empirically, we observe that PT fails in well-specified settings (e.g., Gaussian noise, Figure 5), consistent with our analysis. Our goal is not to characterize all such conditions, but to show that as long as such vulnerabilities exist, coverage–length alone is insufficient.
>
> We are more than pleased to make more clarification for the reviewer's further concerns.

---

> > ### Author Rebuttal · Reviewer_jobk · 2026-04-04
> >
> > Thank you for the detailed rebuttal and the additional experiments. The broader empirical evaluation is helpful and strengthens the paper.
> >
> > That said, my main concerns remain only partially addressed. The new results suggest that the issue is not limited to average set size, but this extension is still mainly empirical, and I still do not see a general characterization of when and why PT appears favorable across different criteria. The connection to practical conformal methods is also clearer, but it still relies mostly on special or degenerate constructions.
> >
> > My main remaining concern is about interval stability. I asked whether it is broadly informative for standard CP methods that are largely deterministic at inference time. The rebuttal mainly appeals to training randomness, which does not fully address this question. I still find the role and general applicability of interval stability unclear beyond explicitly stochastic settings.
> >
> > Overall, the rebuttal improves the paper, but my concerns about generality and broader applicability remain. I will therefore maintain my current score.

---

> > > ### Author Response · Authors · 2026-04-04
> > >
> > > We appreciate the reviewer for the detailed and constructive comments and we are happy that the reviewers find that the broader empirical evaluation is helpful and strengthens the paper. We apologize that there are still some unsolved concerns caused by our unclear expression and hope that the following additional clarifications could help the reviewer's evaluation.
> > >
> > > >this extension is still mainly empirical, and I still do not see a general characterization of when and why PT appears favorable across different criteria.
> > >
> > > We thank the reviewer for this important question and for pointing out the need for a clearer characterization.
> > >
> > > The key mechanism underlying PT is that it induces a **mixture distribution over prediction sets** that a subset of samples is assigned low-information outputs (e.g., null or near-null sets), while the remaining samples receive more conservative intervals.
> > >
> > > Most commonly used efficiency criteria (including average length, set size, and several criteria in [1]) can be written as **expectations over test samples** of a per-instance functional of the prediction set. As a result, their behavior under PT can be understood through the effect of this mixture.
> > >
> > > Concretely, under PT the evaluation metric can be decomposed into two components corresponding to the two subsets. PT appears favorable whenever:
> > >
> > > (1) the criterion decreases sufficiently on the subset with degenerate/small prediction sets, and
> > >
> > > (2) the increase on the remaining subset (due to more conservative intervals) is comparatively mild.
> > >
> > > This provides a general explanation of when and why PT can improve evaluation metrics which is that **whenever the criterion aggregates performance in expectation and does not heavily penalize degenerate predictions**, the mixture construction can yield a lower overall value.
> > >
> > > Our theoretical analysis for interval length (Theorems 3.3–3.8) formalizes this mechanism for a specific functional and the analysis for other criteria is similar given by replacing the length function L with different criteria function. We agree that extending these results to a broader class of criteria is an important direction and thus in the revision, we will clarify that our results illustrate a more general phenomenon driven by this mixture structure.
> > >
> > >
> > >
> > >
> > > >whether interval stability is broadly informative for standard CP methods that are largely deterministic at inference time.
> > >
> > > We thank the reviewer for the insightful question and apologize for the lack of clarity in our previous response.
> > >
> > > We discuss the role of interval stability under three typical scenarios at inference time:
> > >
> > > (1) **Fully deterministic inference.**
> > > If there is no algorithmic randomness at inference time, interval stability is zero by definition, since it is computed based on the variance induced by such randomness. Methods such as CQR (Table 7 in appendix), when implemented deterministically, fall into this category.
> > >
> > > (2) **Largely deterministic inference with minor randomness.**
> > > In some cases, the prediction procedure is mostly deterministic but may involve limited sources of randomness (e.g., tie-breaking in classification settings such as RAPS (Table 8 in appendix)). In this regime, interval stability is expected to be small. However, it is generally difficult to precisely quantify how small stability should be, or to disentangle the contribution of different sources of variability.
> > >
> > > (3) **Randomized inference procedures.**
> > > For CP algorithms with explicit randomness at inference time, interval stability can be large. This is the primary setting studied in our paper.
> > >
> > > For the second scenario, a key challenge is that one can hardly decouple the effect of PT-like mechanisms from other sources of variability. As a result, when comparing methods, it may be difficult to attribute improvements in evaluation metrics to genuine modeling gains versus artifacts induced by such effects.
> > >
> > > In this context, interval stability provides a useful signal: **a non-zero stability indicates the presence of variability in the prediction sets across runs**, which may suggest potential risks in interpretation. While it does not isolate the exact source of this variability, it can serve as an indicator that the observed improvements (e.g., in coverage–length metrics) should be interpreted with caution. Therefore, even in largely deterministic settings, interval stability remains informative as a complementary diagnostic, highlighting potential inconsistencies that are not captured by standard evaluation metrics.
> > >
> > > We thank the reviewer again for providing us the opportunity to reclarify these points, which greatly helps us enhance our manuscripts. We are eager to provide any further clarifications to help your evaluation!

---

### Official Review · Reviewer_8gjF · 2026-03-24

**Soundness:** 2
**Presentation:** 3
**Significance:** 2
**Originality:** 3
**Overall Recommendation:** 3
**Confidence:** 3

**Summary:**

This paper challenges the conventional coverage length metric for evaluating Conformal Prediction Algorithms.The paper introduces an algorithm that uses a Prejudicial Trick ( PT) to construct prediction sets that maintain valid coverage guarantees and outperform in terms of average prediction set length, but the algorithm naively and randomly sets some of the prediction sets to be empty, and uses an adjusted threshold on the rest of the examples to obtain valid coverage. Such an algorithm is not useful in practice due to its inherent randomness and unintuitive prediction sets. Thus, the authors introduce an additional metric called interval stability and argue that this new metric should be evaluated in addition to the coverage-length duo in order to make sure that the conformal algorithm’s gains in terms of length are meaningful and not due to similar randomness like the exemplary  PT algorithm the authors introduced.

**Compliance With Llm Reviewing Policy:**

Affirmed.

**Final Justification:**

I have stated below in the rebuttal acknowledgement that several key concerns remain, as explained below, and thus I am unable to change my assessment to a full accept, but I decided to raise my score as the authors have partially addresses some of my concerns.

**Key Questions For Authors:**

1. Can the authors evaluate the existing CP methods in terms of interval stability?
2. Could the authors clarify the claim and explain it more in line 120, on how for example Conformal Training (stutz et. al ) introduces randomness ? Currently this claim is rather vague. However, there are related works on improving conformal training by reducing the variance of the gradients during the training procedure that lead to improved prediction set lengths  ( Conformal Risk Minimization with Variance Reduction). Would this support your claim, since they show that by reducing the variance there is actually improvement in the prediction set length? Could the authors please comment on this ?
3. Could the authors show how the method of Barber et al (2020) which is directly mentioned in line 192 second column performs in terms of the new interval stability metric?
4. I found the paragraph starting in line 239 rather confusing. The statistics reported are typically those averaged over multiple training runs, and calibration sets which makes this paragraph counterintuitive. Could you please clarify if I am missing something here.

5. Please see weaknesses above

**Limitations:**

Yes

**Strengths And Weaknesses:**

STRENGTHS:
1.  The paper in novel and addresses and highlights an interesting shortcoming of the coverage length metric in conformal prediction.
2. I found the paper well written and the authors explained all theorems intuitively and I found the remarks to be very helpful and well posed at the right points of the paper. This made reading the paper easy and enjoyable.
3. The authors not only highlight a potential weakness of the coverage length metric, but they also address the issue by introducing a new evaluation metric for conformal prediction which closes the loop for their argument.

WEAKNESSES:

I will summarize the main weakness of the paper which led to my decision as well. I believe that if the authors are able to address the following weakness, the paper is comprehensive and the CP community would benefit from the contribution. However, due to the following reasons I currently find the paper not convincing enough or of practical significance for the community:

1. I find the PT algorithm to be rather unique in how it constructs its prediction sets. I understand how this particular trick can improve the length while preserving coverage guarantees. However, this PT algorithm is not well connected to the existing literature in CP. There are high level remarks in the paper of how PT can be connected to Localized CP in line 240 , and line 121 about the break-ties procedures, these remarks are not validated experimentally or shown theoretically. Thus there is no evidence in the paper that any of the existing CP algorithms include a variation or similarity to the PT trick and would then benefit from the new interval stability metric proposed by the authors. A first step could be an experiment where some of the existing CP algorithms are evaluated in terms of the new metric and some insights about the existing well-established methods is revealed. Without this information, it is hard to evaluate the practicality and importance of the new metric and the PT observation for the CP community.

2. Additionally, I would see PT as an extreme case of such unwanted randomness and hence would expect the other algorithm to exhibit milder instability if any compared to  PT algorithm proposed. Yet, in Table 2, the improvements in terms of length seem not statistically significant enough, due to the high variance in length when PT is shown to have smaller length, and the improvements are very small. Out of the 10 datasets shown in this table, there is one only 1/10 ( BIO) where the length is improved and the variance across runs is not significantly higher than the VCP baseline.

---

> ### Author Rebuttal · Authors · 2026-03-29
>
> We thank the reviewer for the positive feedback on novelty, clarity, and importance. We understand the main concern is the practical relevance of PT and its connection to existing CP methods. We clarify that our goal is not to claim that current methods explicitly implement PT, but to show that **the standard coverage–length metric is fundamentally insufficient to distinguish meaningful improvements from pathological ones**. We provide additional experiments at: https://anonymous.4open.science/r/icml2026-rebuttal-8440-AC7E/Reviewer_8gjF.md
>  and address the concerns below.
>
> **1. On the connection to existing CP methods** (Weakness 1).
>
> We agree that PT is a stylized construction. It is designed as a minimal counterexample showing that even with valid marginal and conditional coverage, interval length can be artificially improved. Our claim is not that existing methods explicitly implement PT, **but that uncontrolled randomness can implicitly induce PT-like effects**. In Proposition 3.2, we show that PT can be viewed as a special case of localized CP under a degenerate distribution of the scale estimator $\hat{\sigma}(x)$. Localized CP uses the normalized score $\hat{S}^{\text{norm}}(x,y) = \hat{S}(x,y)/\hat{\sigma}(x)$, where $\hat{\sigma}(x)$ is learned via stochastic procedures (e.g., neural networks or random forests), and is therefore random across runs. When $\hat{\sigma}(x)$ follows a two-point distribution, localized CP reduces exactly to PT.
>
> To bridge this gap, we also consider a milder variant where $\hat{\sigma}(x)=0$ is replaced by a small constant $\epsilon$, which avoids null sets while exhibiting similar behavior (Table A in the link). This makes it closer to practical localized CP settings.
>
> **2. On the empirical significance of PT** (Weakness 2).
>
> The goal of our experiments is not to demonstrate large improvements, but to show that **even small gains in length can be achieved without meaningful improvement in informativeness**. The observed variance across runs reflects instability, which is precisely what interval stability is designed to capture.
>
> Thus, the key point is not the magnitude of improvement, but that **the coverage–length metric alone cannot detect such pathological behavior**. To further address the reviewer’s concern, we include additional experiment showing that under stronger model misspecification (e.g., larger bias), the effect becomes more pronounced (Table B in the link).
>
> **3. On evaluating existing CP methods using interval stability** (Q1).
>
> We thank the reviewer for this suggestion. We have already conducted such evaluations in the appendix: Table 3 (VCP), Table 7 (CQR), and Table 8 (RAPS), where several standard CP methods are evaluated using interval stability. We agree this is important and will move part of these results to the main paper for better visibility.
>
> **4. On conformal training** (Q2).
>
> We thank the reviewer for naming these works. We clarify the distinction between variance in VR-ConfTr and instability in our work.
>
> (1) VR-ConfTr addresses training-time variance. Reducing gradient variance improves optimization, and since the objective includes prediction set size, this leads to shorter intervals via better learning.
>
> (2) In contrast, PT introduces inference-time instability on a fixed model. It reduces average length not by improving uncertainty estimation, but by exploiting randomness (e.g., returning empty sets) to trade per-instance consistency for lower expected length.
>
> **5. On Barber et al (2020)** (Q3).
>
> We note that the construction in Lemma 2 of Barber et al. (2020) is similar to PT, as it also mixes prediction sets with different coverage levels. However, their focus is on conditional coverage guarantees, while our work highlights how such constructions can artificially improve length. In this sense, our work provides a complementary perspective.
>
> **6. On the line 239** (Q4).
>
> We thank the reviewer for pointing out this confusion. The paragraph refers to the scenario where the scale estimator $\hat{\sigma}(x)$ is retrained multiple times, making the resulting interval length a random variable across runs. We agree that in our experiments, results are reported as averages over multiple runs. However, our point is that **averaging over runs does not eliminate variability, but rather hides it**. In particular, while two methods may have similar average length, one may exhibit significantly higher variability across runs for the same input (e.g., due to stochastic training or cherry-picking effects as discussed in Line 239). Such variability is not captured by standard metrics. Therefore, the purpose of this paragraph is to highlight that the coverage-length metric is insufficient to characterize run-to-run variability, which motivates the introduction of interval stability as a complementary metric.
>
> We are more than pleased to make more clarifications if the reviewer has further concerns.

---

> > ### Author Rebuttal · Reviewer_8gjF · 2026-04-07
> >
> > Thank you for the detailed rebuttal. Several concerns remain unresolved.
> > 1. Connection to practical CP methods: Tables 7 and 8 show that the current methods have 0 interval stability which actually further emphasizes the concern that the new metric might not be useful in capturing something non-trivial that current method might exhibit beyond the extreme case of PT. The authors claim PT is theoretically connected to localized CP via Proposition 3.2, so it would be natural to run localized CP and show it achieves shorter sets with higher instability than a baseline or just showing how it performs in terms of the new metric. The theoretical link is appreciated but remains unsubstantiated empirically.
> > 2. The claim regarding Conformal Training (partially unresolved). The training-time vs. inference-time distinction in the rebuttal is helpful, but the original claim in Line 120 that ConfTr-style methods introduce PT-like randomness is still insufficiently supported. What specifically is the randomness being referred to, and is this a theoretical possibility or something empirically observed?
> > 3. On interval stability as a metric (partially unresolved). The paper critiques length as an imperfect proxy for informativeness  but interval stability faces the same limitation. In downstream decision-making settings, what matters is the utility of individual predicted labels, not merely run-to-run consistency of set size. I think the paper makes a valid observation but overstates the significance of interval stability as the appropriate remedy.
> > 4. On overall significance: The core contribution is a well-constructed demonstration that an artificial trick can game the coverage-length metric, and that a new metric can detect it. This is a legitimate and clearly presented observation. However, no existing method has been shown to exhibit this behavior in practice, and the length improvements even under PT are modest. I am open to being convinced otherwise, but the authors need to make a stronger case for the practical relevance of these findings beyond the artificial construction.
> > The following is considered resolved: Q4.
> >
> > **That said, I thank the authors for their response, and the paper was well-written which I appreciate. Given the concerns above, I am not able to change my score to an accept, but I will raise my score as some of my concerns have been addressed in the rebuttal.**

---

> > > ### Author Response · Authors · 2026-04-08
> > >
> > > We thank the reviewer for the detailed follow-up and for raising the score. We would like to further clarify the remaining concerns:
> > >
> > > (1) **On interval stability of existing methods.**
> > >
> > >  We would like to clarify that the interval stability values of RAPS in Table 8 are not zero, but small. In fact, interval stability captures the randomness introduced by procedures such as tie-breaking in RAPS. To further substantiate the practical relevance, we additionally evaluated localized CP and compared it with VCP in terms of interval stability and the results (see Table C in anonymous link https://anonymous.4open.science/r/icml2026-rebuttal-8440-AC7E/Reviewer_8gjF.md) show that localized CP exhibits noticeably higher instability under different rounds of training.The structure of neural net we use to train $\sigma(x)$ is the same as [1].
> > >
> > > [1] Seedat N, Jeffares A, Imrie F, et al. Improving adaptive conformal prediction using self-supervised learning[C]//International Conference on Artificial Intelligence and Statistics. PMLR, 2023: 10160-10177.
> > >
> > > (2) **On conformal training.**
> > >
> > > Our intention in Line 120 was to point out that randomness can arise in the prediction set construction stage, even when using conformal training. Specifically, conformal training methods typically rely on standard CP procedures (e.g., APS) at inference time, where mechanisms such as random tie-breaking introduce stochasticity. We did not claim that this randomness is PT-like, and we will revise the wording to make this distinction explicit.
> > >
> > > (3) **On the role of interval stability.**
> > >
> > > We fully agree that utility in downstream tasks is the ultimate objective. Our goal is not to replace utility-based evaluation, but to complement the standard coverage–length paradigm by exposing run-to-run variability that expectation-based metrics cannot capture. In practice, CP methods are often selected before the downstream task based on reported average performance. Our point is that if such averages can be influenced by randomness, this may lead to unreliable choices in downstream applications. Interval stability is intended as a diagnostic signal to reveal this potential issue.
> > >
> > > (4) **On the magnitude of improvements.**
> > >
> > >  We observe that under stronger model misspecification (e.g., increased bias), the effect becomes more pronounced. We have included additional experimental results in Table B in the anonymous link https://anonymous.4open.science/r/icml2026-rebuttal-8440-AC7E/Reviewer_8gjF.md to illustrate this behavior.
> > >
> > > We sincerely thank the reviewer for this valuable suggestion, which has helped improve the completeness of our paper. We will carefully revise the manuscript based on your comments and are very grateful for your thoughtful feedback.
> > >
> > > # Table C
> > >
> > > | Dataset    | VCP Interval Stability | Localized CP Interval Stability |
> > > | ---------- | ---------------------: | ------------------------------: |
> > > | meps_19    |    0.000000 ± 0.000000 |             0.626625 ± 0.336278 |
> > > | meps_20    |    0.000000 ± 0.000000 |             0.339027 ± 0.273228 |
> > > | meps_21    |    0.000000 ± 0.000000 |             0.895285 ± 0.593803 |
> > > | bike       |    0.000000 ± 0.000000 |             0.027149 ± 0.011657 |
> > > | blog_data  |    0.000000 ± 0.000000 |             1.413026 ± 1.082875 |
> > > | bio        |    0.000000 ± 0.000000 |             0.025216 ± 0.013508 |
> > > | facebook_1 |    0.000000 ± 0.000000 |             0.640953 ± 0.578813 |
> > > | facebook_2 |    0.000000 ± 0.000000 |             0.402816 ± 0.265733 |
> > > | concrete   |    0.000000 ± 0.000000 |             0.112879 ± 0.075699 |
> > > | star       |    0.000000 ± 0.000000 |             0.013176 ± 0.006593 |
> > >
> > > **Caption:** Interval stability comparison between VCP and Localized CP whose instability comes from different rounds of training of $\sigma(x)$.

---

### Decision · Program_Chairs · 2026-04-30

**Decision:**

Accept (regular)

**Comment:**

This paper introduces the Prejudicial Trick (PT), a simple randomized construction that preserves valid coverage while potentially reducing average prediction set size under conditions the paper characterizes, demonstrating that the standard coverage-length (this paper uses average length of prediction sets/intervals as the predictive efficiency measure) paradigm in conformal prediction can be systematically gamed.

All reviewers agree that PT is a clean counterexample. Three reviewers (nx5b, xjbL, mp8c) scored 5 with concerns “Fully resolved”. Two (8gjF, jobk) scored 3 with concerns “Partially resolved or unresolved”. Note that both nx5b and mp8c indicated their scores should be read as between 4 and 5, placing the effective consensus around 4.0.

The core disagreement is practical relevance. Reviewers 8gjF and jobk argue that no existing CP method exhibits PT-like behavior in practice, and that standard methods show zero or near-zero IS in the authors' own experiments. The authors added localized CP results showing non-trivial IS, but as Reviewer jobk noted, this does not establish that localized CP's length advantage is materially driven by the variability IS captures. The gap between "can be gamed in principle" and "is being gamed in practice" remains open.

I recommend weak acceptance because the conceptual contribution, which establishes that coverage and length together may be insufficient to distinguish genuine improvements from pathological ones.

However, the practical value is currently limited significantly: standard CP methods show zero or near-zero IS, and the theoretical results are largely recharacterizations of the core observation, instead of independently useful tools, as Reviewer mp8c (confidence 5) noted. The paper would benefit from more honest positioning as primarily conceptual and from streamlining the theoretical presentation as Reviewer mp8c suggested. As Reviewer jobk noted, the practical case would be substantially strengthened by demonstrating that some CP method in current use exhibits non-trivial IS and that a meaningful portion of its length advantage disappears once variability is controlled for.